# Learning Adaptive Lighting via Channel-Aware Guidance

## Abstract

Learning lighting adaption is a key step in obtaining a good visual perception and supporting downstream vision tasks. There are multiple light-related tasks (e.g., image retouching and exposure correction) and previous studies have mainly investigated these tasks individually. However, we observe that the light-related tasks share fundamental properties: i) different color channels have different light properties, and ii) the channel differences reflected in the time and frequency domains are different. Based on the common light property guidance, we propose a Learning Adaptive Lighting Network (LALNet), a unified framework capable of processing different light-related tasks. Specifically, we introduce the color-separated features that emphasize the light difference of different color channels and combine them with the traditional color-mixed features by Light Guided Attention (LGA). The LGA utilizes color-separated features to guide color-mixed features focusing on channel differences and ensuring visual consistency across channels. We introduce dual domain channel modulation to generate color-separated features and a wavelet followed by a vision state space module to generate color-mixed features. Extensive experiments on four representative light-related tasks demonstrate that LALNet significantly outperforms state-of-the-art methods on benchmark tests and requires fewer computational resources. *We provide an anonymous online demo at https://xxxxxx2025.github.io/LALNet/.*

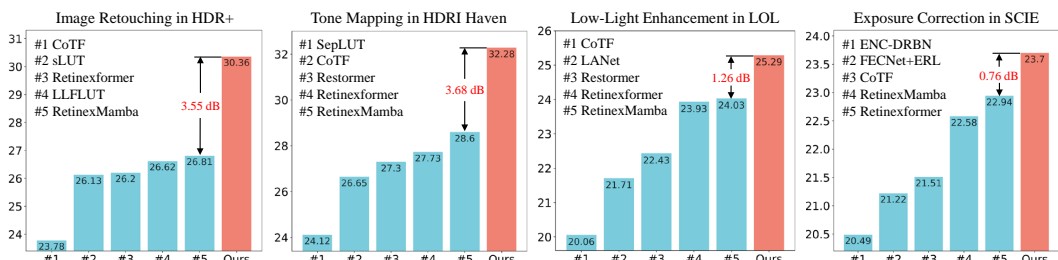

Figure 1: Our LALNet significantly outperforms state-of-the-art methods on four representative benchmark tests of light-related image enhancement, including image retouching, tone mapping, low-light enhancement, and exposure correction.

## 1 Introduction

Photography is the art of light. Images taken under poor lighting conditions often suffer from poor quality, which not only affects image visual presentation but also poses challenges to subsequent computer vision tasks such as target detection and tracking. Therefore, learning adaptive lighting becomes a critical step in obtaining a good visual perception and supporting downstream vision tasks. This process is similar to the perception of the human visual system, that is, light adaptation, which enables us to maintain stable visual perception under diverse lighting environments.

Many tasks in computer vision aim to achieve light adaptation, including image retouching (He et al., 2020; Zhang et al., 2024), tone mapping (Cao et al., 2023; Yang et al., 2022), low-light enhancement (Cai et al., 2023; Bai et al., 2024), and exposure correction (Li et al., 2024a; Huang et al., 2023). The common goal of these light-related tasks is to adjust the light level of the scene to the perceptually optimal level, thereby revealing more visual details. However, due to the different characteristics of these light-related tasks, most of the current methods (Zeng et al., 2020; Li et al., 2024a; Zhang et al., 2019b) are designed to deal with the above tasks individually and are difficult to apply to other light-related tasks. For example, image retouching (Wang et al., 2023; Su et al., 2024) aims to enhance the aesthetic visual quality of images affected by light defects, often requiring special

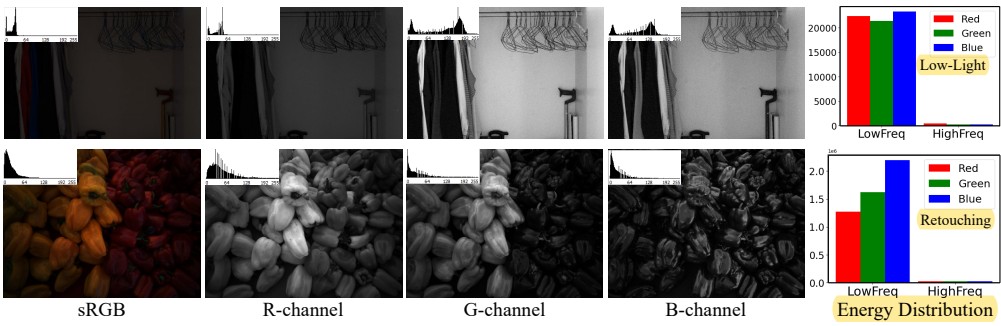

Figure 2: Motivation of our method. Visualization of different color channel differences and statistical DWT spectral energy distributions of different tasks.

attention to global light; tone mapping (Zhang et al., 2022; Wang et al., 2021) preserves rich details by compressing high dynamic range light to low dynamic range, focusing more on adaptation to high dynamic range light; low-light enhancement (Wang et al., 2022; Liu et al., 2021a) reveals more details by boosting the brightness of dark areas, but requires special processing of noise; and exposure correction (Huang et al., 2022b; Zhang et al., 2019b) must adjust the brightness of both underexposed and overexposed scenes to achieve clearer images. The different characteristics of these tasks make existing methods inconsistent in performance on multiple tasks. Although some works (Yang et al., 2023a; Zhang et al., 2021) have attempted to perform light-related tasks with a unified architecture, the insufficient analysis of light-related task specificity has resulted in unsatisfactory performance compared to methods designed for these individual tasks.

*Interestingly, can a unified framework be designed to handle these light-related tasks, just as the human visual system can adapt to a variety of lighting environments?* Motivated by this question, we aim to design a unified framework capable of handling multiple light enhancement tasks separately.

To this end, we delve deep into analyzing the common light properties of these light-related tasks and utilize them to inspire the design of our unified framework. We observe two key insights from light-related tasks: **i) different color channels have different light properties**; **ii) the channel differences reflected in the time and frequency domains are different.** To analyze these differences, we employ the Discrete Wavelet Transform (Shensa et al., 1992) to decompose the input image into low-frequency and high-frequency components, and statistics on the energy distribution of the R/G/B channels based on the square of the pixel values separately. Fig. 2 illustrates the color channel attributes of two light-related task images in the time and frequency domains. It can be observed that the light properties of different channels differ significantly and that there is no fixed pattern between the different images. For example, for the first image, the G-channel exhibits a more balanced luminance distribution, while for the second image, the R-channel performs better in this regard. On the other hand, the frequency domain exhibits channel differences that are different from the time domain. For example, in the first image, the G-channel is brighter, but the R-channel has the highest energy distribution in the frequency domain. This illustrates that capturing channel differences in the time and frequency domains is different. Channel differences cannot be fully characterized in the time or frequency domains alone. More analysis is provided in the appendix. Moreover, it is well known that the specific attributes (Yang et al., 2023a; Liang et al., 2021b; Zhang et al., 2024) of light-related tasks are mainly embodied in the low-frequency components, whereas the details of the contents are more related to the high-frequency components. These findings highlight the importance of learning adaptive lighting by leveraging distinctive features of different color channels in the time and frequency domains.

Motivated by the above light properties, we propose a unified light adaptation framework, namely LALNet. Our method leverages the potential channel light differences to guide effective adaptive lighting. We decompose the light adaptation problem into two sub-tasks: (i) light adaptation, which addresses light variations under different light conditions, and (ii) detail enhancement, which preserves and refines image details while performing adaptive lighting. We begin to learn adaptive light enhancement from downsampled low-resolution images, optimizing for low computational complexity. To implement light adaptation, we propose a dual-branch architecture comprising channel separation and channel mixing. The channel separation branch employs the Dual Domain Channel Modulation (DDCM) module to extract color-separated features, focusing on light differences and color-specific luminance distributions for each channel in the frequency and time domains. In the

channel mixing branch, we apply wavelet feature modulation and vision state space module to integrate color-mixed lighting information, capturing inter-channel relationships and lighting patterns that achieve balanced light enhancement. A key component of our framework is Light Guided Attention (LGA), which utilizes color-separated features to guide color-mixed light information for adaptive lighting. This mechanism enhances the network's capability to perceive changes in channel luminance differences and ensure visual consistency and color balance across channels. Consequently, our network is effectively adaptive to light variations while attending to feature differences across channels. Finally, we employ an iterative detail enhancement strategy to recover the image resolution level by level while enhancing the details. We conduct comprehensive experiments and demonstrate the state-of-the-art performance of our LALNet on four light-related tasks, as shown in Fig. 1. Our contributions can be summarized as follows:

- In this paper, we propose a unified light adaptation framework inspired by the common light property, namely the Learning Adaptive Lighting Network (LALNet).
- We introduce the Dual Domain Channel Modulation to capture the light differences of different color channels and combine them with the traditional color-mixed features by Light Guided Attention.
- Extensive experiments on four representative light-related tasks show that LALNet significantly outperforms state-of-the-art methods in benchmarking and that our method requires fewer computational resources.

## 2 METHODS

### 2.1 MOTIVATION

Previous studies (Cai et al., 2023; Li et al., 2024a; Zhang et al., 2024; Su et al., 2024) for light-related tasks, such as tone mapping and low-light enhancement, are often tailored to individual tasks, leading to suboptimal performance across multiple scenarios. These frameworks typically fail to account for the common properties shared across different lighting-related tasks, which limits their generalizability. As a result, many frameworks are either overly specialized or inefficient when faced with multiple tasks. This leads to performance inconsistencies, especially when frameworks designed for specific tasks are applied to others. For instance, Retinexformer focuses on separating reflection and illumination to enhance low-light images, but its underlying Retinex theory is inapplicable to tasks such as tone mapping and image retouching. This limitation is evident in scenarios where low-light enhancement methods struggle to maintain color fidelity during tone mapping. Our motivation is rooted in the observation that, despite the diverse nature of light-related tasks, there are key shared properties: **distinct light properties across color channels** and **channel differences in time and frequency domains.** These channel differences manifest differently in both the time and frequency domains, further complicating the task of adaptive lighting. To address these issues, we aim to design a unified framework that adapts to different lighting conditions more effectively than previous frameworks that focus on individual tasks. By analyzing these shared light properties across multiple tasks, our framework seeks to capture the subtle differences between color channels and ensure consistent and balanced visual outcomes across various lighting conditions.

### 2.2 FRAMEWORK OVERVIEW

The overall pipeline of LALNet is illustrated in Fig. 3. Our framework is composed of two key components: light adaptation and detail enhancement. Given a low-quality (LQ) input image $\mathbf{X}$, our goal is to generate a high-quality (HQ) output $\mathbf{Y}$ with optimal light. We begin to learn adaptive light enhancement from downsampled low-resolution images $\mathbf{X}_{LF}^3$, optimizing for low computational complexity. Subsequently, we employ the two-branch structure for extracting light features, containing color separation and color mixing branches. The channel separation branch employs the DDCM and group convolution modules to extract color-separated feature $\mathbf{F}_{cs}$, focusing on light differences and color-specific luminance distributions for each channel in the time and frequency domains. In the channel mixing branch, we utilize wavelet feature modulation combined with the vision state space module (VSSM) to extract color-mixed feature $\mathbf{F}_{cm}$, promoting cross-channel interaction and achieving balanced light enhancement. This can be expressed mathematically as:

$$\mathbf{F}_{cs} = \text{GConv}(\text{DDCM}(\mathbf{X}_{LF}^3)), \quad \mathbf{F}_{cm} = \text{VSSM}(\text{WFM}(\mathbf{X}_{LF}^3)). \tag{1}$$

To emphasize the light differences in different channels, we introduce Light Guided Attention, which injects the color-separated features into color-mixed features to obtain the light adaptive feature $F_{la}$,

Figure 3: Architecture of LALNet for light adaptation. The core modules of LALNet are: (a) dual domain channel modulation (DDCM) that extracts color-separated features, focusing on light differences for each channel in the frequency and time domains, and (b) light guided attention (LGA) utilizes color-separated features to guide color-mixed light information for light adaptation.

which is described as:

$$\mathbf{F}_{\text{la}} = \text{LGA}(\mathbf{F}_{\text{cm}}, \mathbf{F}_{\text{cs}}). \tag{2}$$

This process ensures consistent and uniform light adaptation across the entire image and eliminates color distortion caused by channel crosstalk. Finally, we integrate the low- and high-frequency components via learnable differential pyramid and iterative detail enhancement, progressively refining image resolution and enhancing fine details.

## 2.3 LIGHT ADAPTATION

In the literature, we generally utilize the traditional convolutions to convolve with all channels for light-related tasks, generating RGB-mixed features. This operator can capture the interaction information and shared features among channels. However, this also amplifies the luminance non-uniformity and noise existing in the three channels. Notably, for light-related tasks, we have observed that characteristic differences between the RGB channels and the time and frequency domains exhibit different differences. There is also no consistent pattern across images. As shown in Fig. 2, the three channels exhibit distinct differences in luminance, with one channel usually being closer to ground truth. If we only utilize color-mixed features to adapt to light, the negative interference between channels will also spread to all channels. Therefore, we introduce an additional branch that extracts channel-separated features alongside the channel-mixed features. Channel-mixed features are responsible for capturing mixed luminance and color information, while channel-separated features guide the network to focus on channel differences. This design prompts the network to adapt to light while attending to feature differences across channels.

### 2.3.1 COLOR SEPARATION REPRESENTATION

Based on the analysis in Sec. 1, the time and frequency domains reflect different channel differences. Therefore, we employ DDCM to capture the color-separated features.

**Dual Domain Channel Modulation.** To avoid cross-channel interference between operating channels in the spatial domain, we process each channel independently in the frequency and time domains and introduce learnable parameters to modulate the channels. After frequency domain processing, the images are inverted back to the time domain. Then, to complement the color-separated feature representation, we utilize channel attention to capture the color-separated features in the time domain.

Specifically, given an input image $\mathbf{X}$, each channel of the image is denoted as $\mathbf{X}_i$ ($i = 1, 2, 3$). We perform a 2D fast Fourier Transform (FFT) for $\mathbf{X}_i$ to obtain the frequency domain representation:

$$\mathbf{S}_i(u, v) = \mathcal{F}(\mathbf{X}_i)(u, v) = \text{FFT2}(\mathbf{X}_i), \tag{3}$$

where $\mathbf{S}_i(u, v) = \mathbf{R}_i(u, v) + j \cdot \mathbf{I}_i(u, v)$, $\mathbf{R}_i(u, v)$ and $\mathbf{I}_i(u, v)$ denote the real and imaginary parts, respectively. Then, we perform convolution operations on the $\mathbf{R}_i(u, v)$ and $\mathbf{I}_i(u, v)$, respectively:

$$\hat{\mathbf{R}}_i(u, v) = \mathbf{W}_{R_i} * \mathbf{R}_i(u, v), \quad \hat{\mathbf{I}}_i(u, v) = \mathbf{W}_{I_i} * \mathbf{I}_i(u, v), \tag{4}$$

where $\mathbf{W}_{R_i}$ and $\mathbf{W}_{I_i}$ are the convolution kernels, $*$ denote convolution operation. Afterward, we predict weight for the $\hat{\mathbf{R}}_i(u, v)$ and $\hat{\mathbf{I}}_i(u, v)$ and apply the weights to the real and imaginary parts

after convolution:

$$\mathbf{\Lambda}_{R_i} = \text{softmax}(\hat{\mathbf{W}}_{R_i} * \hat{\mathbf{R}}_i), \quad \mathbf{\Lambda}_{I_i} = \text{softmax}(\hat{\mathbf{W}}_{I_i} * \hat{\mathbf{I}}_i), \tag{5}$$

$$\mathbf{R}'_i(u, v) = \hat{\mathbf{R}}_i(u, v)\mathbf{\Lambda}_{R_i}, \quad \mathbf{I}'_i(u, v) = \hat{\mathbf{I}}_i(u, v)\mathbf{\Lambda}_{I_i}, \tag{6}$$

where $\hat{\mathbf{W}}_{R_i}$ and $\hat{\mathbf{W}}_{I_i}$ are the convolution weights, $\text{softmax}$ denote the activation function. Subsequently, we reorganize the decoupled real and imaginary parts into frequency-domain signals, and perform the Inverse Fourier Transform to obtain the decoupled time-domain information as follows:

$$\mathbf{S}'_i(u, v) = \mathbf{R}'_i(u, v) + j \cdot \mathbf{I}'_i(u, v), \tag{7}$$

$$\mathbf{X}'_i = \mathcal{F}^{-1}(\mathbf{S}'_i(u, v)) = \text{IFFT2}(\mathbf{S}'_i). \tag{8}$$

Finally, after concatenating channels, we capture the separated features of image in the time domain through the channel attention module to further enhance the color-separated feature representation.

$$\mathbf{F}_{\text{cs}} = \text{CAB}(\text{Concat}(\mathbf{X}'_1, \mathbf{X}'_2, \mathbf{X}'_3)). \tag{9}$$

### 2.3.2 COLOR MIXING REPRESENTATION

In parallel, we introduce wavelet feature modulation for extracting channel-mixed features. Since light patterns often exhibit global characteristics (Rieke & Rudd, 2009; Yang et al., 2023a), inspired by (Finder et al., 2024), we employ wavelet transform to achieve channel-mixed features $F_{\text{cm}}$. The process begins with the extraction of small-scale features using a small convolutional kernel to capture local information. These features are then passed through a Wavelet Transform Block (WTB), where the generated large-scale features modulate the small-scale features, enabling the network to better integrate global light representation. The process can be represented as follows:

$$\mathbf{cA}, \mathbf{cH}, \mathbf{cV}, \mathbf{cD} = \text{WTB}(\text{Conv}_{3\times3}(\mathbf{X})), \tag{10}$$

Afterward, the modulated features are concatenated and further passed the convolutional layer.

$$\mathbf{F}_{\text{cm}} = \text{Conv}_{3\times3}(\text{Concat}(\mathbf{cA}, \mathbf{cH}, \mathbf{cV}, \mathbf{cD})). \tag{11}$$

To further enhance the network's ability to capture global light information, we complement wavelet feature modulation with the vision state space module (Guo et al., 2024). This module can efficiently capture long-range dependencies without being computationally expensive as in transformer-based methods. Specifically, VSSM first extends the channel to $2C$ by a linear layer and then splits it into two features according to the channel dimensions, which serve as inputs to two parallel branches. In the first branch, the channels are expanded to $\eta C$ using a linear layer, followed by depth-wise convolution, SiLU activation, 2D selective scanning, and LayerNorm. 2D selective scanning transforms 2D image features into linear sequences by scanning in four orientations: top-left to bottom-right, bottom-right to top-left, top-right to bottom-left, and bottom-left to top-right. Each sequence's dependencies are modeled using discrete state-space equations, and the outputs from all sequences are merged and reshaped back into a 2D format. The second branch directly activates the original features via SiLU. Finally, the outputs of both branches are multiplied and compressed back to the original dimensions using a linear layer. The whole process can be represented as follows:

$$\mathbf{F}_1, \mathbf{F}_2 = \text{Chunk}(\text{Linear}(\mathbf{F}_{\text{cm}})), \tag{12}$$

$$\mathbf{F}'_1 = \text{LN}(\text{SS2D}(\text{SiLU}(\text{DWConv}(\mathbf{F}_1)))), \quad \mathbf{F}'_2 = \text{SiLU}(\mathbf{F}_2), \tag{13}$$

$$\mathbf{F}^1_{\text{cm}} = \text{MLP}(\text{LN}((\mathbf{F}'_1 \otimes \mathbf{F}'_2))), \tag{14}$$

where $\text{Linear}(\cdot)$ denote linear projection, $\otimes$ denotes the Hadamard product.

### 2.3.3 LIGHT GUIDED ATTENTION

Although VSSM performs well in capturing long-range dependencies, it still faces problems such as local information forgetting and channel redundancy. Moreover, color mixed features ignore the feature differences between different channels, treating them equally in the network. However, in light-related tasks, we have observed significant differences between color channels, with no consistent pattern across images. These differences are crucial for adaptive lighting. For this reason, we propose to inject color-separated features into color-mixed features by light guided attention to perceive channel differences.

Specifically, for first LGA module, we input the channel-mixed features $\mathbf{F}_{cm}^1$ from VSSM and the channel-separated features $\mathbf{F}_{cs}^1$ from group convolution into the LGA. Subsequently, the input $\mathbf{F}_{cm}^1$ is processed through a $1 \times 1$ convolution followed by a depthwise convolution, producing $\mathbf{K}$ and $\mathbf{V}$ tensor with doubled the number of channels. This can be expressed mathematically as:

$$\mathbf{K}, \mathbf{V} = \text{Conv}_{3\times3}(\text{Conv}_{1\times1}(\mathbf{F}_{cm}^1)). \tag{15}$$

The query $\mathbf{Q}$ is then generated from the channel-separated features $\mathbf{F}_{cs}^1$:

$$\mathbf{Q} = \text{Conv}_{3\times3}(\text{Conv}_{1\times1}(\text{GConv}_{3\times3}(\mathbf{F}_{cs}^1))). \tag{16}$$

We compute the attention weights by the dot product between $\mathbf{Q}$ and $\mathbf{K}$, normalized by the softmax function, and multiplied by $\mathbf{V}$ to obtain the updated features:

$$\text{Attention}(\mathbf{Q}, \mathbf{K}, \mathbf{V}) = \text{softmax}(\frac{\mathbf{Q}\mathbf{K}^T}{\sqrt{d_K}} \times \tau)\mathbf{V}, \tag{17}$$

where $d_K$ is the dimension of $\mathbf{K}$ and $\tau$ denotes the scaling factor. It can be remarked that we utilize channel-separated features as $\mathbf{Q}$ vectors to motivate the model to focus on channel differences. In summary, the design of LGA enhances the adaptive representation of image features in both spatial and channel dimensions and improves the network's ability to capture dependencies between image channels. After LGA processing, we can obtain the low-resolution light-adaption output $\mathbf{Y}_{LF}^L$. Subsequently, we utilize the iterative detail enhancement strategy to enhance the detail of $\mathbf{Y}_{LF}^L$, which is introduced in the following.

## 2.4 DETAIL ENHANCEMENT

To achieve faithful reconstruction, we apply a learnable differential pyramid (LDP) to capture high-frequency details. Through LDP, we obtain the complete multi-scale high-frequency features $\mathbb{X}_{HF} = [\mathbf{X}_{HF}^0, \ldots, \mathbf{X}_{HF}^{L-1}]$, tapering resolutions from $H \times W$ to $\frac{H}{2^{L-1}} \times \frac{W}{2^{L-1}}$. $L$ denotes the number of pyramid levels ($L=3$ in our framework). More details about the implementation of LDP are provided in the appendix.

Using the high-frequency information $\mathbb{X}_{HF}$ captured by the LDP, we employ an iterative detail enhancement to progressively refine the light-adaption image $\mathbf{Y}_{LF}^L$. Specifically, for the $l_{th}$ pyramid, we first up-sample the low-frequency image $\mathbf{Y}_{LF}^l$ and concatenate it with HF component $\mathbf{X}_{HF}^{l-1}$, then feed it into a residual network to predict a refinement mask $\mathbf{M}^{l-1}$. This mask allows pixel-by-pixel refinement of the HF component, which is subsequently added to the up-sampling $\mathbf{Y}_{LF}^l$ to generate the reconstructed result of the current layer $\mathbf{Y}_{LF}^{l-1}$. The process at the $l_{th}$ pyramid is formulated as:

$$\mathbf{M}^{l-1} = \text{Res}(\text{Concat}(\text{Up}(\mathbf{Y}_{LF}^l), \mathbf{X}_{HF}^{l-1})), \qquad \mathbf{Y}_{LF}^{l-1} = \text{Up}(\mathbf{Y}_{LF}^l) + (\mathbf{X}_{HF}^{l-1}\mathbf{M}^{l-1}), \tag{18}$$

where $\text{Res}(\cdot)$ and $\text{Up}(\cdot)$ denote the residual block and up-sampling, respectively.

## 2.5 LOSS FUNCTIONS

We utilize three objective losses to optimize our network, including reconstruction loss, perceptual loss, and high-frequency loss.

**Reconstruction loss.** To maintain the accuracy of the reconstructed image, we directly adopt pixel-wise $L_{Re}$ and $L_{SSIM}$ loss on the final prediction $\mathbf{Y}$ and the ground truth $\mathbf{G}$:

$$L_{Re} = \sum_{l=0}^{L} \left\| \mathbf{Y}_{LF}^l - \mathbf{G}_{LF}^l \right\|_1, \tag{19}$$

$$L_{SSIM} = 1 - \text{SSIM}(\mathbf{Y}, \mathbf{G}), \tag{20}$$

where $\mathbf{Y}_{LF}^l$ denotes the output of each layer of the network and $\mathbf{G}_{LF}^l$ denotes the Gaussian pyramid of the ground truth.

**High-frequency loss.** To efficiently reconstruct high-frequency details, we introduce a high-frequency loss function. By calculating the $L_1$ loss between the output high-frequency component and the high-frequency of ground truth:

$$L_{HF} = \sum_{l=0}^{L-1} \left\| \mathbf{Y}_{HF}^l - \mathbf{G}_{HF}^l \right\|_1, \tag{21}$$

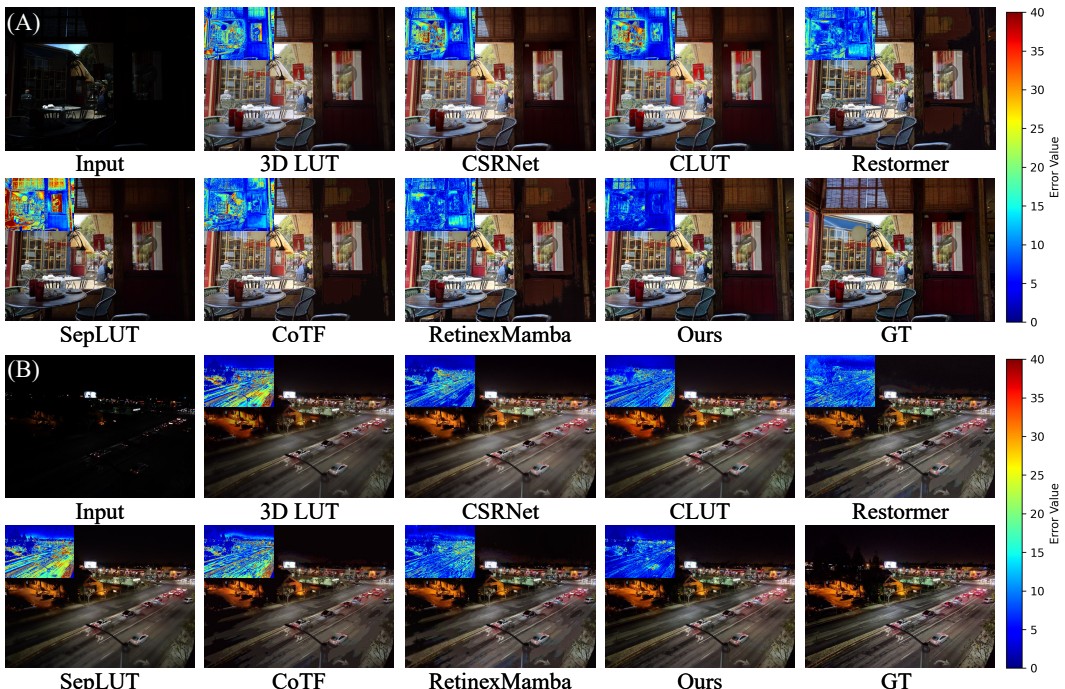

Figure 4: Visual comparisons between our LALet and the state-of-the-art methods on the HDR+ dataset (Zoom-in for best view). The error maps in the upper left corner facilitate a more precise determination of performance differences.

where $\mathbf{G}_{\mathrm{HF}}^{l}$ denotes the HF component of the ground truth obtained through the Laplacian pyramid.
**Perceptual loss.** To obtain more robust adaptive light, we employ a perceptual loss function that assesses a solution concerning perceptually relevant characteristics (e.g., the structural contents and detailed textures):

$$L_{\mathrm{P}} = \mathrm{VGGLoss}(\mathbf{Y}, \mathbf{G}), \tag{22}$$

where VGGLoss represents the 5-th convolution layer within VGG19 network (Simonyan, 2015).
**Output loss.** To summarize, the complete objective of our proposed model is combined as follows:

$$L_{\mathrm{total}} = \alpha \cdot L_{\mathrm{Re}} + \beta \cdot L_{\mathrm{SSIM}} + \gamma \cdot L_{\mathrm{HF}} + \eta \cdot L_{\mathrm{P}}, \tag{23}$$

where $\alpha$, $\beta$, $\gamma$, and $\eta$ are the corresponding weight coefficients.

## 3 EXPERIMENTS

### 3.1 EXPERIMENTAL SETTINGS

**Datasets.** We evaluate our method on four representative light-related tasks: image retouching (HDR+ Burst Photography (Hasinoff et al., 2016)), tone mapping (HDRI Haven [1], exposure correction (SCIE (Cai et al., 2018)), low-light enhancement (LOL dataset (Wei et al., 2018)). The HDR+ dataset is a staple for image retouching, especially in mobile photography. We utilize 675 image sets for training and 248 for testing. The HDRI Haven dataset is widely recognized as one of the benchmarks for evaluating tone mapping (Cao et al., 2023; Su et al., 2021), which includes 570 HDR images of diverse scenes under various light conditions. We select 456 image sets for training and 114 for testing. Following the settings of (Huang et al., 2022a) for SICE, it contains 1000 training images, and 24 test images. LOL dataset (Wei et al., 2018) contains 500 image pairs in total, with 485 pairs used for training and 15 pairs set aside for testing.

**Implementation details.** We implement our model with Pytorch on the NVIDIA L40s GPU platform. The model is trained with the Adam optimizer ($\beta_1 = 0.9$, $\beta_2 = 0.999$) for $4 \times 10^5$ iterations. The learning rate is initially set to $2 \times 10^{-4}$ and then steadily decreased to $1 \times 10^{-6}$ by the cosine annealing scheme during the training process. We adopt traditional PSNR and SSIM metrics on the

---

[1]https://hdri-haven.com/

Table 1: Quantitative results of image retouching and tone mapping methods. "/" denotes the unavailable source code. Metrics with ↑ and ↓ denote higher better and lower better. The best and second results are in red and blue, respectively.

| Method | #Params | Image Retouching in HDRPlus | | | | | | |
|---|---|---|---|---|---|---|---|---|
| | | PSNR↑ | SSIM↑ | TMQI↑ | LPIPS↓ | △E↓ | NIQE↓ | MUSIQ↑ |
| UPE (Wang et al., 2019a) | 999K | 23.33 | 0.852 | 0.856 | 0.150 | 7.68 | 12.75 | 66.98 |
| HDRNet (Gharbi et al., 2017) | 482K | 24.15 | 0.845 | 0.877 | 0.110 | 7.15 | 10.47 | 68.73 |
| CSRNet (He et al., 2020) | 37K | 23.72 | 0.864 | 0.884 | 0.104 | 6.67 | 10.99 | 67.82 |
| DeepLPF (Moran et al., 2020) | 1.72M | 25.73 | 0.902 | 0.877 | 0.073 | 6.05 | 10.35 | 70.02 |
| LUT (Zeng et al., 2020) | 592K | 23.29 | 0.855 | 0.882 | 0.117 | 7.16 | 11.36 | 67.67 |
| CLUT (Zhang et al., 2022) | 952K | 26.05 | 0.892 | 0.886 | 0.088 | 5.57 | 11.19 | 67.39 |
| LPTN (Liang et al., 2021b) | 616K | 24.80 | 0.884 | 0.885 | 0.087 | 8.38 | 12.44 | 67.99 |
| sLUT (Wang et al., 2021) | 4.52M | 26.13 | 0.901 | / | 0.069 | 5.34 | / | / |
| SepLUT (Yang et al., 2022) | 120K | 22.71 | 0.833 | 0.879 | 0.093 | 8.62 | 12.26 | 67.89 |
| Restormer (Zamir et al., 2022) | 26.1M | 25.93 | 0.900 | 0.883 | 0.050 | 6.59 | 10.49 | 68.92 |
| LLFLUT (Zhang et al., 2024) | 731K | 26.62 | 0.907 | / | 0.063 | 5.31 | / | / |
| CoTF (Li et al., 2024a) | 310K | 23.78 | 0.882 | 0.876 | 0.072 | 7.76 | 11.54 | 68.07 |
| Retinexformer (Cai et al., 2023) | 1.61M | 26.20 | 0.910 | 0.879 | 0.046 | 6.14 | 10.75 | 68.93 |
| RetinexMamba (Bai et al., 2024) | 4.59M | 26.81 | 0.911 | 0.880 | 0.047 | 5.89 | 10.52 | 69.02 |
| LALNet-Tiny | 246K | 29.68 | 0.939 | 0.882 | 0.031 | 4.81 | 9.78 | 70.07 |
| LALNet-Lite | 536K | 30.09 | 0.945 | 0.886 | 0.028 | 4.52 | 9.81 | **70.31** |
| LALNet | 2.87M | **30.36** | **0.946** | **0.888** | **0.026** | **4.48** | 9.87 | 70.29 |

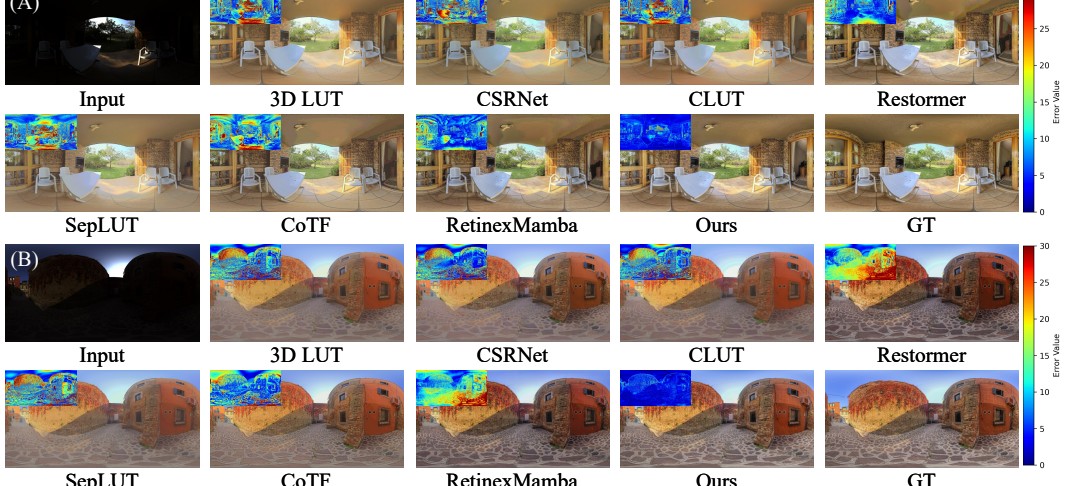

Figure 5: Visual comparisons between our LALet and the state-of-the-art methods on the HDRI Haven dataset (Zoom-in for best view). The error maps in the upper left corner facilitate a more precise determination of performance differences.

RGB channel to evaluate the reconstruction accuracy. We also employ TMQI (Yeganeh & Wang, 2013), LPIPS (Zhang et al., 2018) and CIELAB color space (Zhang et al., 1996) to evaluate image quality and perceptual quality respectively.

## 3.2 COMPARISON WITH STATE-OF-THE-ARTS

**Quantitative comparison.** The performance of the proposed unified framework is evaluated on four light-related image enhancement tasks, namely, (1) image retouching, (2) tone mapping, (3) exposure correction, and (4) low-light enhancement. We quantitatively compare the proposed method with a wide range of state-of-the-art light-related methods in Tab. 1, Tab. 2, and Appendix. For image retouching, as shown in Tab. 1, the proposed LALNet outperforms all the previous SOTA methods by a large margin. Specifically, our method significantly outperforms the SOTA methods RetinexFormer (Cai et al., 2023), LLFLUT (Zhang et al., 2024) and CoTF (Li et al., 2024a), RetinexMamba (Bai et al., 2024), improving PSNR by **3.55 dB** in the HDR+ dataset. Notably, our LALNet-Tiny has only $246K$ parameters and $1.62G$ FLOPs, but the performance is also significantly better than other SOTA methods. For tone mapping, Tab. 7 reports the quantitative results on the HDRI Haven dataset. We can see that our method has the best overall performance. Our method

Table 2: Quantitative results of exposure correction methods on the SCIE dataset. "/" denotes the unavailable source code.

| Method | Under | | Over | | Average | | | | |
|---|---|---|---|---|---|---|---|---|---|
| | PSNR↑ | SSIM↑ | PSNR↑ | SSIM↑ | PSNR↑ | SSIM↑ | LPIPS↓ | NIQE↓ | MUSIQ↑ |
| URtinexNet (Wu et al., 2022) | 17.39 | 0.6448 | 7.40 | 0.4543 | 12.40 | 0.5496 | 0.3549 | 12.78 | 49.11 |
| DRBN (Yang et al., 2020) | 17.96 | 0.6767 | 17.33 | 0.6828 | 17.65 | 0.6798 | 0.3891 | 12.06 | 48.77 |
| SID (Chen et al., 2018) | 19.51 | 0.6635 | 16.79 | 0.6444 | 18.15 | 0.6540 | 0.2417 | 11.79 | 51.07 |
| MSEC (Afifi et al., 2021) | 19.62 | 0.6512 | 17.59 | 0.6560 | 18.58 | 0.6536 | 0.2814 | / | / |
| SID-ENC (Huang et al., 2022a) | 21.30 | 0.6645 | 19.63 | 0.6941 | 20.47 | 0.6793 | 0.2797 | 11.49 | 52.29 |
| DRBN-ENC (Huang et al., 2022a) | 21.89 | 0.7071 | 19.09 | 0.7229 | 20.49 | 0.7150 | 0.2318 | 11.23 | 54.15 |
| CSRNet (He et al., 2020) | 21.43 | 0.6789 | 20.13 | 0.7250 | 20.78 | 0.7019 | 0.1390 | 10.59 | 61.79 |
| CLIP-LIT (Liang et al., 2023) | 15.13 | 0.5847 | 7.52 | 0.4383 | 11.33 | 0.5115 | 0.3560 | / | / |
| FECNet (Huang et al., 2022b) | 22.01 | 0.6737 | 19.91 | 0.6961 | 20.96 | 0.6849 | 0.2656 | 11.05 | 53.73 |
| FECNet+ERL (Huang et al., 2023) | 22.35 | 0.6671 | 20.10 | 0.6891 | 21.22 | 0.6781 | / | / | / |
| CoTF (Yang et al., 2023a) | 22.90 | 0.7029 | 20.13 | 0.7274 | 21.51 | 0.7151 | 0.1924 | 10.19 | 51.61 |
| Retinexformer (Cai et al., 2023) | 23.75 | 0.7157 | 22.13 | 0.7466 | 22.94 | 0.7310 | 0.1714 | 10.37 | 55.67 |
| RetinexMamba (Bai et al., 2024) | 23.56 | 0.7212 | 21.59 | 0.7384 | 22.58 | 0.7298 | 0.1856 | 10.35 | 53.67 |
| LALNet-Tiny | 23.77 | 0.7135 | 22.01 | 0.7484 | 22.89 | 0.7310 | 0.1258 | 9.56 | 62.94 |
| LALNet | 24.55 | 0.7291 | 22.85 | 0.7596 | 23.70 | 0.7444 | 0.1327 | 9.53 | 62.42 |

Table 3: Ablation studies of key components.

| Variants | WFM | DDCM | LGA | PSNR↑ | SSIM↑ |
|---|---|---|---|---|---|
| #1 | ✓ | ✗ | ✗ | 29.11 | 0.933 |
| #2 | ✓ | ✓ | ✗ | 29.58 | 0.935 |
| #3 | ✓ | ✗ | ✓ | 30.01 | 0.942 |
| #4 | ✗ | ✓ | ✓ | 30.05 | 0.942 |
| #5 | ✓ | ✓ | ✓ | 30.36 | 0.946 |

Table 4: Ablation studies on different loss functions on the HDR+ dataset.

| Variants | $L_{Re}$ | $L_{HF}$ | $L_{SSIM}$ | $L_p$ | PSNR↑ | SSIM↑ |
|---|---|---|---|---|---|---|
| #1 | ✓ | ✗ | ✓ | ✓ | 30.14 | 0.944 |
| #2 | ✓ | ✓ | ✗ | ✓ | 29.88 | 0.941 |
| #3 | ✓ | ✓ | ✓ | ✗ | 29.72 | 0.939 |
| #4 | ✓ | ✓ | ✓ | ✓ | 30.36 | 0.946 |

has the best performance with 32.28 dB PSNR, 0.969 SSIM, 0.961 TMQI, 0.019 LPIPS, and 3.69 $\Delta E$. For exposure correction, Tab. 2 report the quantitative results on the SCIE. As can be seen, our method improves **1.19 dB** PSNR and 0.0293 SSIM compared to the CoTF (Li et al., 2024a) (CVPR24) method. For low-light enhancement, Our LALNet significantly outperforms SOTA methods on the LOL-v1 dataset while requiring moderate computational and memory costs. Compared with the recent best method RetinexMamba (Bai et al., 2024), LALNet achieves **1.26 dB** PSNR and 0.027 SSIM. However, our method only costs 16% (6.86 / 42.82) GFLOPs.

**Qualitative results.** Visual comparison of LALNet and state-of-the-art light-related image enhancement methods are shown in Fig. 4, Fig. 5, Fig. 6, and Fig. 10. Please zoom in for better visualization. To better visualize the performance differences of various methods, we present an error map to show the differences between the results of each method and the target image, as shown in the upper left corner of the image. In the error map, the red area indicates a larger difference, while the blue area indicates that the two are closer. It is worth noting that error maps have no special units and only indicate errors. These figures illustrate that our LALNet consistently delivers visually appealing results on light-related tasks. Results reveal the proposed method usually obtains better precise color reconstruction and vivid color saturation. Meanwhile, our method faithfully reconstructs fine high-frequency textures. For instance, in Fig. 4, our method exhibits excellent color fidelity and restores proper global brightness and local contrast, consistent colors, and sharp details. In Figure 5, the second best method, RetinexMamba, exhibits ghosting and dead blacks, but our LALNet still performs well. These results prove that our method produces more pleasing visual effects. More results and visual comparisons are presented in our Supplementary Material.

## 3.3 ABLATION STUDIES

We conduct comprehensive breakdown ablations to evaluate the effects of our proposed framework.

**Effectiveness of specific modules.** To validate the effectiveness of the DDCM, WFM, and LGA modules in the low-frequency pathway, we set up different variants to validate the effectiveness of the proposed framework. The results are listed in Tab. 3. Variants #1 is removing the color-separated branch, with a performance drop of 1.25 dB. For Variants #2, we remove the LGA module and directly sum channel-mixed and channel-separated features for light guidance. The results confirm the effectiveness of the color-separated feature to guide the light adaptation, with a PSNR increase of

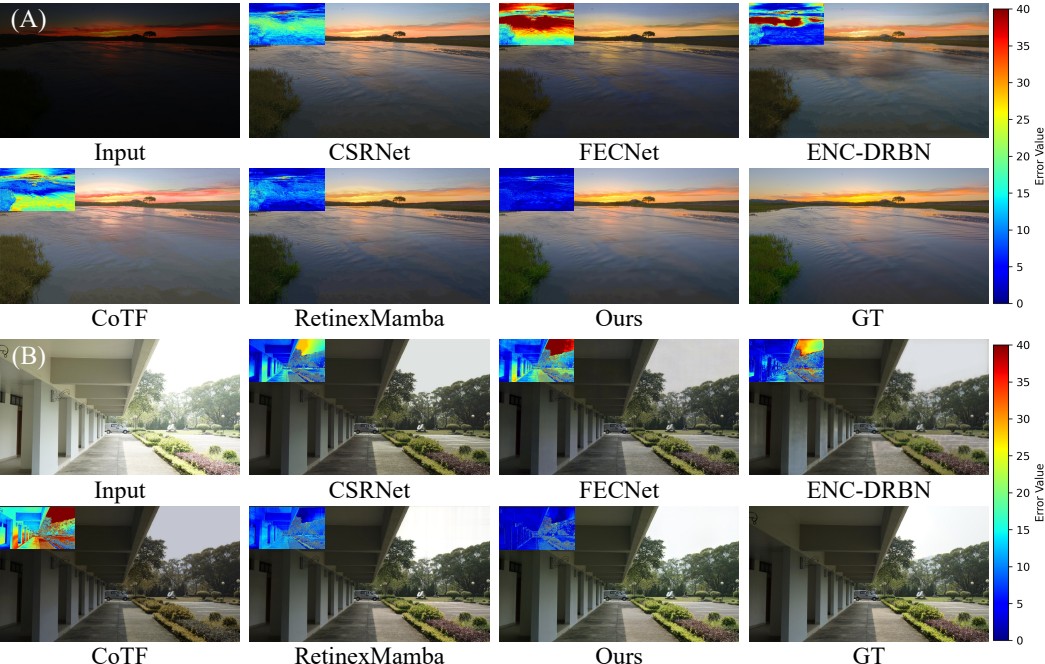

Figure 6: Visual comparisons between our LALet and the SOTA methods on the SCIE dataset.

0.47 dB. In Variants #3, we use group convolution replacing DDCM to extract channel-separated features, and the PSNR is reduced by 0.35 dB. Similarly, Variants #4 apply a convolution block to replace the WFM with a performance reduction of 0.31 dB PSNR. The results show that our proposed DDCM and WFM are effective compared to conventional feature extraction. These results consistently demonstrate the effectiveness of our method.

**Ablation study on loss functions.** To test the effect of the loss function on the performance, we set up different variants and modified the loss function combination step by step. Tab. 4 shows that adding $L_p$ or $L_{SSIM}$ loss can improve performance. In particular, the addition of $L_p$ loss results in 0.64 PSNR higher than the baseline. Meanwhile, $L_{HF}$ is equally positive for the performance gain.

**Selection of the number of levels.** We validate the influence of the number of pyramid levels $l$. As shown in Tab. 5, the model achieves the best performance on all tested resolutions when $l = 3$. When a larger number of levels ($l \geq 4$) result in a significant decline in performance. This is because when $l$ is larger and the number of downsamples is more, the model fails to reconstruct the high frequencies efficiently, resulting in performance degradation. When $l = 1$, the low-frequency image resolution equals the input image resolution, leading to a burst of computational memory. Comparing $l = 2$ and $l = 3$ demonstrates that despite the small input image resolution of the low-frequency pathway, high-frequency details can still be recovered efficiently in our framework.

Table 5: Ablation study on the pyramid levels number. The "N.A." result is not available due to insufficient GPU memory.

| Metrics | Number of Levels | | | |
|---|---|---|---|---|
| | n=1 | n=2 | n=3 | n=4 |
| PSNR | N.A. | 30.25 | **30.36** | 29.23 |
| SSIM | N.A. | 0.943 | **0.946** | 0.936 |
| TMQI | N.A. | 0.879 | **0.887** | 0.879 |
| LPIPS | N.A. | 0.029 | **0.026** | 0.031 |
| △E | N.A. | 4.75 | **4.49** | 5.00 |
| #Params | **2.62M** | 2.71M | 2.87M | 3.23M |
| FLOPs | 38.71G | 12.84G | 6.86G | **5.54G** |

## 4 CONCLUSION

This paper proposes a unified framework for learning adaptive lighting via light property guidance. In particular, we propose DDCM for extracting color-separated features and capturing the light difference across channels. The LGA utilizes color-separated features to guide color-mixed features for adaptive lighting, achieving color consistency and color balance. Extensive experiments demonstrate that our method significantly outperforms state-of-the-art methods, improving PSNR by 3.55 dB in the HDR+ dataset, 3.68 dB in the HDRI Haven dataset, 1.26 dB in the SCIE dataset, and 0.76 dB in the LOL dataset respectively compared with the second best method.

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

# Appendix

CONTENTS

In this Appendix, we present the related work and provide additional results and analysis.

# A    RELATED WORK

**Image Retouching** Recently learning-based methods have utilized CNNs (Moran et al., 2020; Liang et al., 2021a; Li et al., 2020; Gao & Wu, 2021) for image retouching, particularly on datasets like MIT-Adobe FiveK (Bychkovsky et al., 2011) and HDR+ (Hasinoff et al., 2016). Some methods (Kim et al., 2020; Li et al., 2020) reformulate retouching images as a curve estimation task. For instance, DeepLPF (Moran et al., 2020) optimizes local filters to achieve fine-grained adjustments. Considering inference time and memory consumption, 3D Lookup Tables (LUTs) (Zeng et al., 2020; Liang et al., 2021a) have been proposed, offering efficient retouching with competitive results. He *et al.* (He et al., 2020) developed CSRNet for efficient image retouching. In addition, GAN-based models (Chen et al., 2018; Ni et al., 2020) have been explored for unpaired supervision.

**Tone Mapping** Learning-based methods have been applied to tone mapping, aiming to bridge the gap between HDR and LDR imaging (Zhang et al., 2022; Yang et al., 2022; Zhang et al., 2024; Hu et al., 2022). CNN-based models (Hou et al., 2017) laid the groundwork for tone mapping, with later works exploring GANs for pixel-level accuracy (Cao et al., 2020; Rana et al., 2020; Panetta et al., 2021). Despite these advancements, issues such as halo artifacts and local inconsistencies persist. Hu *et al.* (Hu et al., 2022) addressed these in a hybrid way, combining tone mapping and denoising using discrete cosine transforms, while Zhang *et al.* (Zhang et al., 2019a) leveraged HSV color space manipulation to reduce halos and enhance detail retention. However, for tasks such as exposure correction and low-light enhancement that require luminance and high-frequency information, the luminance (e.g., L or V channels) is obtained through nonlinear transformations, which may result in loss of or distortion of luminance details. Despite notable progress, existing methods often struggle to balance global and local tone mapping, resulting in unsatisfactory results in other tasks.

**Exposure Correction** Exposure correction tackles the challenge of balancing light in images. Methods like RetinexNet (Liu et al., 2021a) decompose illumination and reflectance for separate enhancement, while ZeroDCE (Guo et al., 2020) uses high-order pixel curves for underexposed images. DRBN (Yang et al., 2020) learns pixel mappings to decompose and recombine images under perceptual guidance. However, these methods primarily focus on underexposure, neglecting the variety of real-world exposure scenarios. Afifi *et al.* (Afifi et al., 2021) introduced a multi-scale Laplacian pyramid network to address diverse exposure challenges, while Huang *et al.* (Huang et al., 2022b) leveraged a Fourier-based network to enable complementary interactions between spatial and frequency domains. More recently, Li *et al.* (Li et al., 2024b) proposed a collaborative transformation framework for real-time exposure correction, efficiently combining global and pixel-level adjustments.

**Low-light Image Enhancement** Deep learning-based methods, particularly CNNs (Yang et al., 2023b; Zhou et al., 2022; Liu et al., 2021b), have made significant strides in low-light enhancement. Wang *et al.* (Wang et al., 2019b) introduced DeepUPE, a Retinex-inspired model for illumination prediction. Xu *et al.* (Xu et al., 2022) developed SNR-Net, a CNN-Transformer hybrid, achieving SOTA performance at the cost of computational efficiency. To mitigate this, Zamir *et al.* introduced Restormer (Zamir et al., 2022), an efficient model with long-range pixel interactions. Cai *et al.* (Cai et al., 2023) extended this further with Retinexformer, setting new benchmarks. Bai *et al.* (Bai et al., 2024) employed State Space Models for computational efficiency in low-light enhancement. However, Retinex-based methods (Cai et al., 2023; Liu et al., 2021a; Bai et al., 2024) are based on the theory of separated illumination and reflection, but they usually assume smooth and uniform lighting conditions, which may not hold in realistic scenes involving complex lighting variations. In addition, these methods typically work in luminance or reflection space, where high-frequency details may be distorted during decomposition. On the other hand, balancing global receptive fields with computational demands remains a core challenge for real-world applicability.

# B    FURTHER ANALYSIS OF MOTIVATION

Different wavelengths of light exhibit different response characteristics when an image sensor captures photons for photoelectric conversion. After processing by an image signal processor, these differential responses are sometimes amplified or minimized but are difficult to eliminate. In addition,

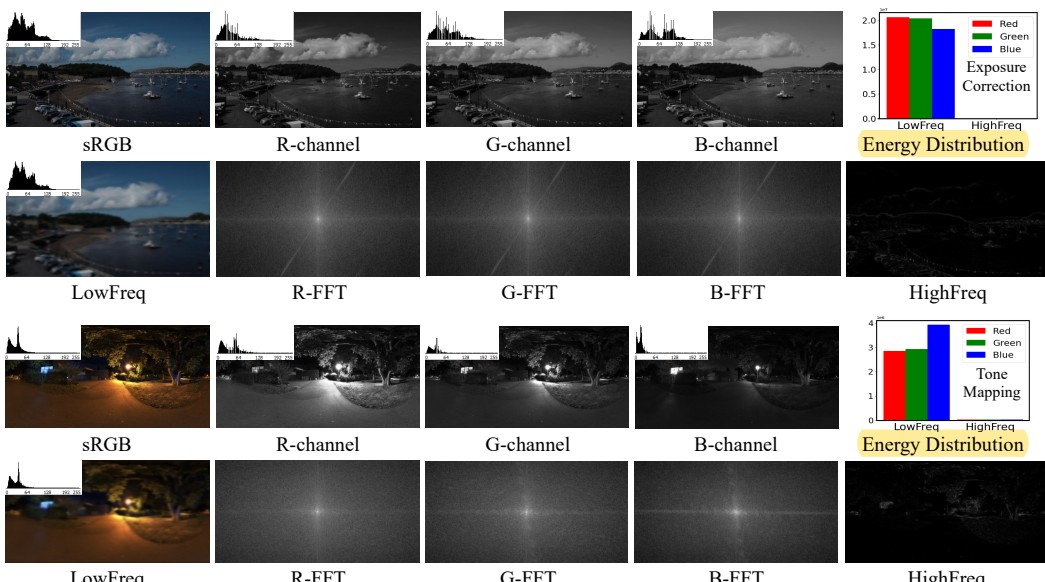

Figure 7: Motivation. Visualization of the light-related task images in different color channels and their corresponding DWT spectra energy distribution. R-FFT denotes the Fourier Frequency Domain diagram of the R channel. LowFreq and HighFreq are low-frequency and high-frequency images.

the differences in the Bayer pattern of different image sensors also result in different channels showing different responses to luminance and noise. Meanwhile, light sources in natural scenes are usually non-uniform, which also leads to the fact that sunlight, shadows, reflections, and other factors can cause RGB channels to respond differently to the same scene.

Recall that in Sec. 1, we discussed two observations that serve as the motivation to design our network. We show more motivation cases in Fig. 7 (From observations of the exposure correction and tone mapping tasks). In particular, (a) different color channels have different light properties, and (b) the channel differences reflected in the time and frequency domains are different. To further analyze our first motivation, we visualized the frequency domain images of the different channels using the Fourier Transform and compared them. The results show that, as in the time domain, significant differences are exhibited between the different channels in the frequency domain. Based on the observations in Fig. 2 and Fig. 7, the common properties of several light-related tasks investigated in this paper are verified, which also contribute to the design of our network.

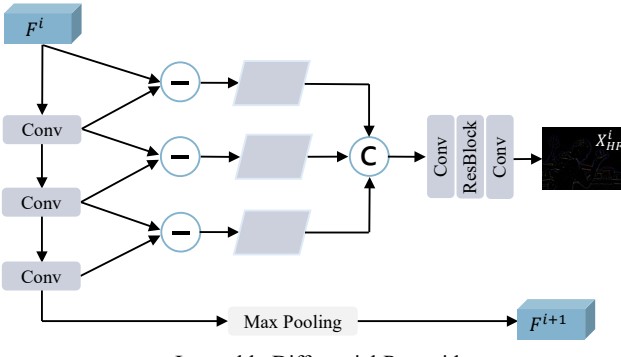

Learnable Differential Pyramid

Figure 8: The architecture of the Learnable Differential Pyramid module that extracts high-frequency information from the input image.

## C  VISUALIZATION IN THE NETWORK

We demonstrate the Learnable Difference Pyramid and Iterative Detail Enhancement modules in Fig. 8 and Fig. 9. To efficiently capture high-frequency details of the input image, inspired by the traditional difference pyramid, we construct a learnable difference pyramid using simple convolution and residual blocks.

In detail, input image $X$ is first processed by an initial convolution to obtain the initial feature map $\mathbf{F}^0$. For each pyramid level $l$, we generate the Gaussian feature map $F^l$ and the high-frequency feature map $F^l_{hf}$ of the current level through the difference module, where the difference module is composed of three successive convolution and maximum pooling operations. The high-frequency feature $\mathbf{F}^l_{hf}$ further generates the high-frequency output $\mathbf{X}^l_{HF} \in \mathbb{R}^{H \times W \times 3}$ through the residual block and Gaussian features $\mathbf{F}^l$ are used as inputs to the next layer. Through $l - 1$ iterations, we obtain the complete differential pyramid $\mathbf{X}_{HF} = [X^0_{HF}, \ldots, \mathbf{X}^{l-1}_{HF}]$ that contains multi-scale high-frequency features adaptively learned from the LQ images, tapering resolutions from $H \times W$ to $\frac{H}{2^{l-1}} \times \frac{W}{2^{l-1}}$.

Meanwhile, in order to reduce the computational resources, we implement light adaptation at low resolution. To compensate for the loss of details, we use an iterative detail enhancement module to recover high-frequency details. Specifically, we first up-sample the low-frequency mapped image $\mathbf{Y}^l_{LF}$ and concatenate it with HF component $\mathbf{X}^{l-1}_{HF}$, then fed it into a residual network to predict the mask $\mathbf{M}_{l-1}$. This mask allows pixel-by-pixel refinement of the HF component, which is subsequently added to the up-sampling $\mathbf{Y}^l_{LF}$ to generate the reconstructed result of the current layer $\mathbf{Y}^{l-1}_{LF}$. The operations at the $l - th$ layer can be formulated as:

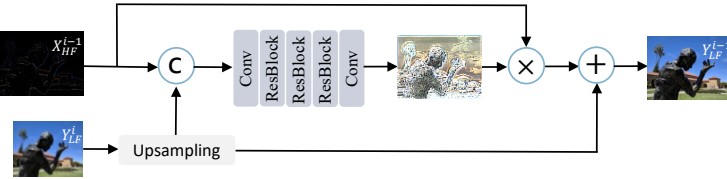

Iterative Detail Enhancement

Figure 9: The architecture of the Iterative Detail Enhancement module progressively restores resolution and fine details.

## D  MORE RESULTS ON RELEASED MODELS

We also further validate the effectiveness of our model in low-light enhancement (Wei et al., 2018), exposure correction (Afifi et al., 2021), HDR Survey (Fairchild, 2023), and UVTM (Cao et al., 2023) datasets that contains more complex lighting. The MSEC dataset (Afifi et al., 2021) renders images using relative EVs of -1.5 to +1.5 and contains a total of 17675 training images, 750 validation images, and 5905 test images. Table 9 reports the quantitative results of the MSCE. We can see that our method has the best overall performance. On the MSEC dataset, our method has the best performance with 23.93 dB PSNR, 0.8734 SSIM, and 0.0791 LPIPS. We validate our model on non-homologous third-party image and video HDR datasets, as shown in Table 6, and our model far outperforms existing methods. The HDR Survey dataset consists of 105 HDR images, with no ground truth, and is one of the benchmarks for HDR tone mapping evaluations (Cao et al., 2020; Rana et al., 2020; Panetta et al., 2021; Liang et al., 2018; Paris et al., 2011). The UVTM video dataset, also with no ground truth, includes 20 real captured HDR videos. Note that the HDR Survey and UVTM video datasets are only for testing purposes.

Table 6: Validating generalization on third-party datasets include HDR Survey and UVTM video datasets.

| Datasets | Metrics | HDRNet | CSRNet | 3D LUT | CLUT | SepLUT | IVTMNet | CoTF | Ours |
|---|---|---|---|---|---|---|---|---|---|
| HDR Survey | TMQI | 0.8641 | 0.8439 | 0.8165 | 0.8140 | 0.8085 | 0.9160 | 0.8612 | **0.9292** |
| UVTM | TMQI | 0.8281 | 0.8973 | 0.8787 | 0.8799 | 0.8629 | 0.8991 | 0.9006 | 0.9576 |

Table 7: Quantitative results of tone mapping methods. "/" denotes the unavailable source code. Metrics with ↑ and ↓ denote higher better and lower better. The best and second results are in red and blue, respectively.

| Method | #Params | Tone Mapping in HDRI Haven | | | | |
|---|---|---|---|---|---|---|
| | | PSNR↑ | SSIM↑ | TMQI↑ | LPIPS↓ | △E↓ |
| UPE (Wang et al., 2019a) | 999K | 23.58 | 0.821 | 0.917 | 0.191 | 10.85 |
| HDRNet (Gharbi et al., 2017) | 482K | 25.33 | 0.912 | 0.941 | 0.113 | 7.03 |
| CSRNet (He et al., 2020) | 37K | 25.78 | 0.872 | 0.928 | 0.153 | 6.09 |
| DeepLPF (Moran et al., 2020) | 1.72M | 24.86 | 0.939 | 0.948 | 0.077 | 7.64 |
| LUT (Zeng et al., 2020) | 592K | 24.52 | 0.846 | 0.912 | 0.171 | 7.33 |
| CLUT (Zhang et al., 2022) | 952K | 24.29 | 0.836 | 0.908 | 0.169 | 7.08 |
| LPTN (Liang et al., 2021b) | 616K | 26.21 | 0.941 | 0.954 | 0.113 | 8.82 |
| SepLUT (Yang et al., 2022) | 120K | 24.12 | 0.854 | 0.915 | 0.165 | 8.03 |
| Restormer (Zamir et al., 2022) | 26.1M | 27.30 | 0.954 | 0.948 | 0.032 | 5.67 |
| CoTF (Li et al., 2024a) | 310K | 26.65 | 0.935 | 0.948 | 0.098 | 5.84 |
| Retinexformer (Cai et al., 2023) | 1.61M | 27.73 | 0.955 | 0.949 | 0.030 | 5.41 |
| RetinexMamba (Bai et al., 2024) | 4.59M | 28.60 | 0.955 | 0.953 | 0.032 | 5.12 |
| LALNet-Tiny | 246K | 31.58 | 0.963 | 0.954 | 0.024 | 4.07 |
| LALNet-Lite | 536K | 31.79 | 0.964 | 0.954 | 0.023 | 3.87 |
| LALNet | 2.87M | **32.28** | **0.969** | **0.961** | **0.019** | **3.69** |

Table 8: Quantitative results of LLE methods on the LOLv1 dataset. "*" denotes that the results are from reference papers.

| Method | FLOPs(G) | Low-Light Enhancement | |
|---|---|---|---|
| | | PSNR↑ | SSIM↑ |
| 3DLUT (Zeng et al., 2020) | 0.075 | 14.35 | 0.445 |
| DeepUPE (Wang et al., 2019b) | 21.10 | 14.38 | 0.446 |
| DeepLPF (Moran et al., 2020) | 5.86 | 15.28 | 0.473 |
| UFormer (Wang et al., 2022) | 12.00 | 16.36 | 0.771 |
| RentinexNet (Wei et al., 2018) | 587.47 | 17.19 | 0.589 |
| EnGAN (Jiang et al., 2021) | 61.01 | 17.48 | 0.650 |
| Sparse (Yang et al., 2021) | 53.26 | 17.20 | 0.640 |
| FIDE (Xu et al., 2020) | 28.51 | 18.27 | 0.665 |
| KinD (Zhang et al., 2019b) | 34.99 | 20.35 | 0.813 |
| CSRNet (He et al., 2020) | 6.6 | 20.46 | 0.659 |
| MIRNet (Zamir et al., 2020) | 785 | 24.14 | 0.842 |
| LANet (Yang et al., 2023a) | / | 21.71 | 0.810 |
| Restormer (Zamir et al., 2022) | 144.25 | 22.43 | 0.823 |
| CoTF (Li et al., 2024a) | 1.81 | 20.06 | 0.755 |
| Retinexformer (Cai et al., 2023)* | 15.57 | 23.93 | 0.831 |
| RetinexMamba (Bai et al., 2024) | 42.82 | 24.03 | 0.827 |
| LALNet-Tiny | 1.62 | 24.06 | 0.845 |
| LALNet | 6.86 | **25.29** | **0.854** |

# E  ABLATION STUDY

To validate the effectiveness of the SS2D module, we use Self-Attention and Residual Block to replace the SS2D module in the original published model. We use the Self-Attention module released by Restormer (Zamir et al., 2022), and ResBlock is constructed from two convolutional layers and activation functions. The results, as shown in Table 10, show that using SS2D as part of the base module effectively captures global features and strikes a balance between performance and efficiency. Notably, the same excellent results are obtained using the Self-Attention module, which is attributed to the design of our overall framework, further demonstrating the effectiveness of our proposed adaptive lighting framework.

Table 9: Quantitative results of exposure correction methods on the MSCE dataset.

| Method | Exposure Correction in MSCE | | | | | | |
|---|---|---|---|---|---|---|---|
| | Under | | Over | | Average | | |
| | PSNR↑ | SSIM↑ | PSNR↑ | SSIM↑ | PSNR↑ | SSIM↑ | LPIPS↓ |
| He (Pizer et al., 1987) | 16.52 | 0.6918 | 16.53 | 0.6991 | 16.53 | 0.6959 | 0.2920 |
| CLAHE (Reza, 2004) | 16.77 | 0.6211 | 14.45 | 0.5842 | 15.38 | 0.5990 | 0.4744 |
| LIME (Guo et al., 2016) | 13.98 | 0.6630 | 9.88 | 0.5700 | 11.52 | 0.6070 | 0.2758 |
| WVM (Fu et al., 2016) | 18.67 | 0.7280 | 12.75 | 0.645 | 15.12 | 0.6780 | 0.2284 |
| RetinexNet (Wei et al., 2018) | 12.13 | 0.6209 | 10.47 | 0.5953 | 11.14 | 0.6048 | 0.3209 |
| URtinexNet (Wu et al., 2022) | 13.85 | 0.7371 | 9.81 | 0.6733 | 11.42 | 0.6988 | 0.2858 |
| DRBN (Yang et al., 2020) | 19.74 | 0.8290 | 19.37 | 0.8321 | 19.52 | 0.8309 | 0.2795 |
| SID (Chen et al., 2018) | 19.37 | 0.8103 | 18.83 | 0.8055 | 19.04 | 0.8074 | 0.1862 |
| MSEC (Afifi et al., 2021) | 20.52 | 0.8129 | 19.79 | 0.8156 | 20.08 | 0.8145 | 0.1721 |
| SID-ENC (Huang et al., 2022a) | 22.59 | 0.8423 | 22.36 | 0.8519 | 22.45 | 0.8481 | 0.1827 |
| DRBN-ENC (Huang et al., 2022a) | 22.72 | 0.8544 | 22.11 | 0.8521 | 22.35 | 0.8530 | 0.1724 |
| CLIP-LIT (Liang et al., 2023) | 17.79 | 0.7611 | 12.02 | 0.6894 | 14.32 | 0.7181 | 0.2506 |
| FECNet (Huang et al., 2022b) | 22.96 | 0.8598 | 23.22 | 0.8748 | 23.12 | 0.8688 | 0.1419 |
| LCDPNet (Zhang et al., 2019b) | 22.35 | 0.8650 | 22.17 | 0.8476 | 22.30 | 0.8552 | 0.1451 |
| FECNet+ERL (Zamir et al., 2020) | 23.10 | 0.8639 | 23.18 | 0.8759 | 23.15 | 0.8711 | / |
| CoTF (Yang et al., 2023a) | 23.36 | 0.8630 | 23.49 | 0.8793 | 23.44 | 0.8728 | 0.1232 |
| LALNet | **23.81** | **0.8636** | **24.05** | **0.8798** | **23.93** | **0.8734** | **0.0791** |

Table 10: Ablation study on the global feature extraction modules.

| Variants | Replaced Modules | #Params | FLOPs | PSNR↑ | SSIM↑ | TMQI↑ | LPIPS↓ | △E↓ |
|---|---|---|---|---|---|---|---|---|
| #1 | ResBlock | 2.99M | 7.13G | 29.77 | 0.9412 | 0.8781 | 0.0291 | 4.760 |
| #2 | Self-Attention | 2.25M | 6.48G | 29.91 | 0.938 | 0.8801 | 0.0297 | 4.872 |
| #3 | Ours | 2.87M | 6.86G | **30.36** | **0.9458** | **0.8883** | **0.0261** | **4.483** |

Further, we use FDCM to capture color-separated features, and to avoid channel mixing during information propagation, we use group convolution to keep the color channels separated. To verify the effectiveness of the design, we use traditional convolution to replace group convolution. The experimental results are shown in Table 11, where the channel mixing caused by the conventional convolution leads to a performance degradation of 0.41 dB. This phenomenon shows the necessity of color channel separation and the effectiveness of using color-separated features to guide light adaptation.

Table 11: Ablation study on the Group Convolution (G-Conv) and traditional Convolution (T-Conv).

| Variants | Replaced Modules | #Params | FLOPs | PSNR↑ | SSIM↑ | TMQI↑ | LPIPS↓ | △E↓ |
|---|---|---|---|---|---|---|---|---|
| #1 | T-Conv | 2.93M | 6.91G | 29.95 | 0.9399 | 0.8791 | 0.0292 | 4.645 |
| #2 | G-Conv | 2.87M | 6.86G | **30.36** | **0.9458** | **0.8883** | **0.0261** | **4.483** |

# F  MORE VISUAL COMPARISON

We present more comparisons between state of the arts for enhancement light-related images in Figures 12, 13, 14, 15, 16, and 17. This is similar to Fig. 4 of the main paper where we compare methods using their original released models. As shown, all existing models do not handle these lighting-related images well. Although RetinexFormer and RetinexMamba obtained the second-best quantitative results in most tasks, the qualitative results show that they suffer from varying degrees of artifacts, which seriously impact the visual quality. This phenomenon also indicates that Retinex-based methods are inapplicable to challenging light tasks.

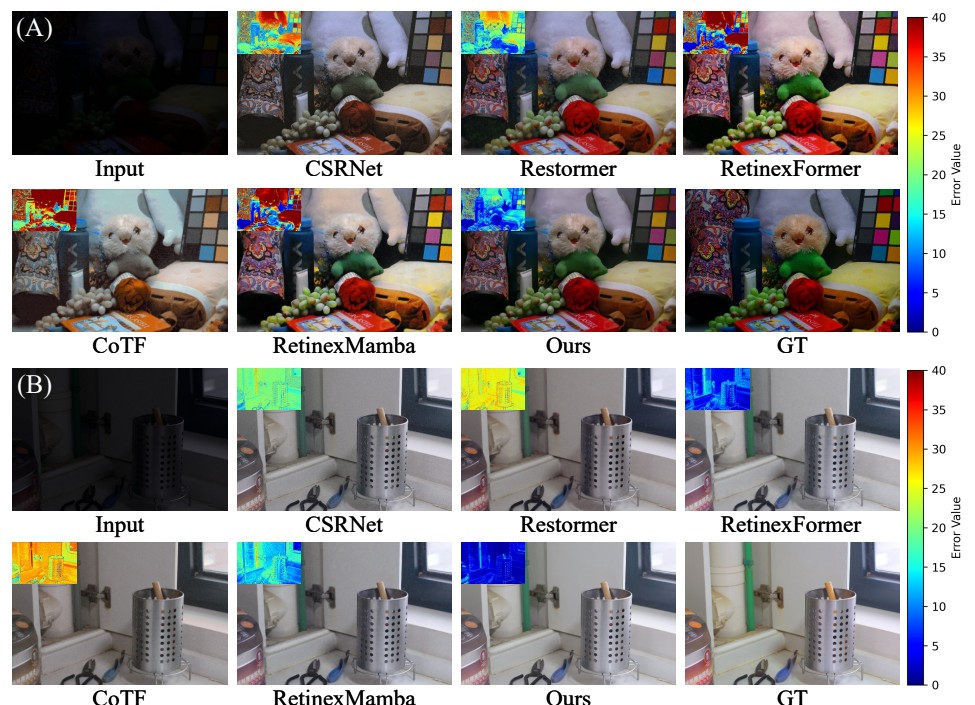

Figure 10: Visual comparisons between our LALet and the SOTA methods on the LOLv1 dataset.

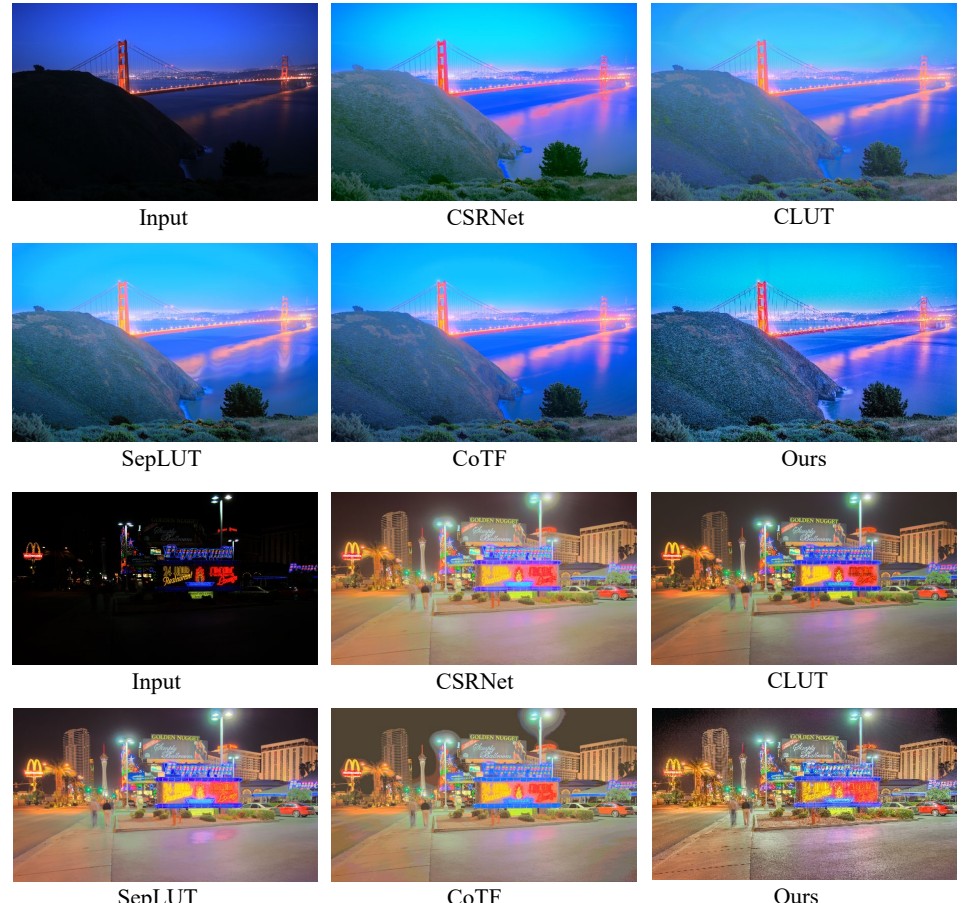

Figure 11: Visual comparisons between our LALet and the SOTA methods on the HDR Survey dataset.

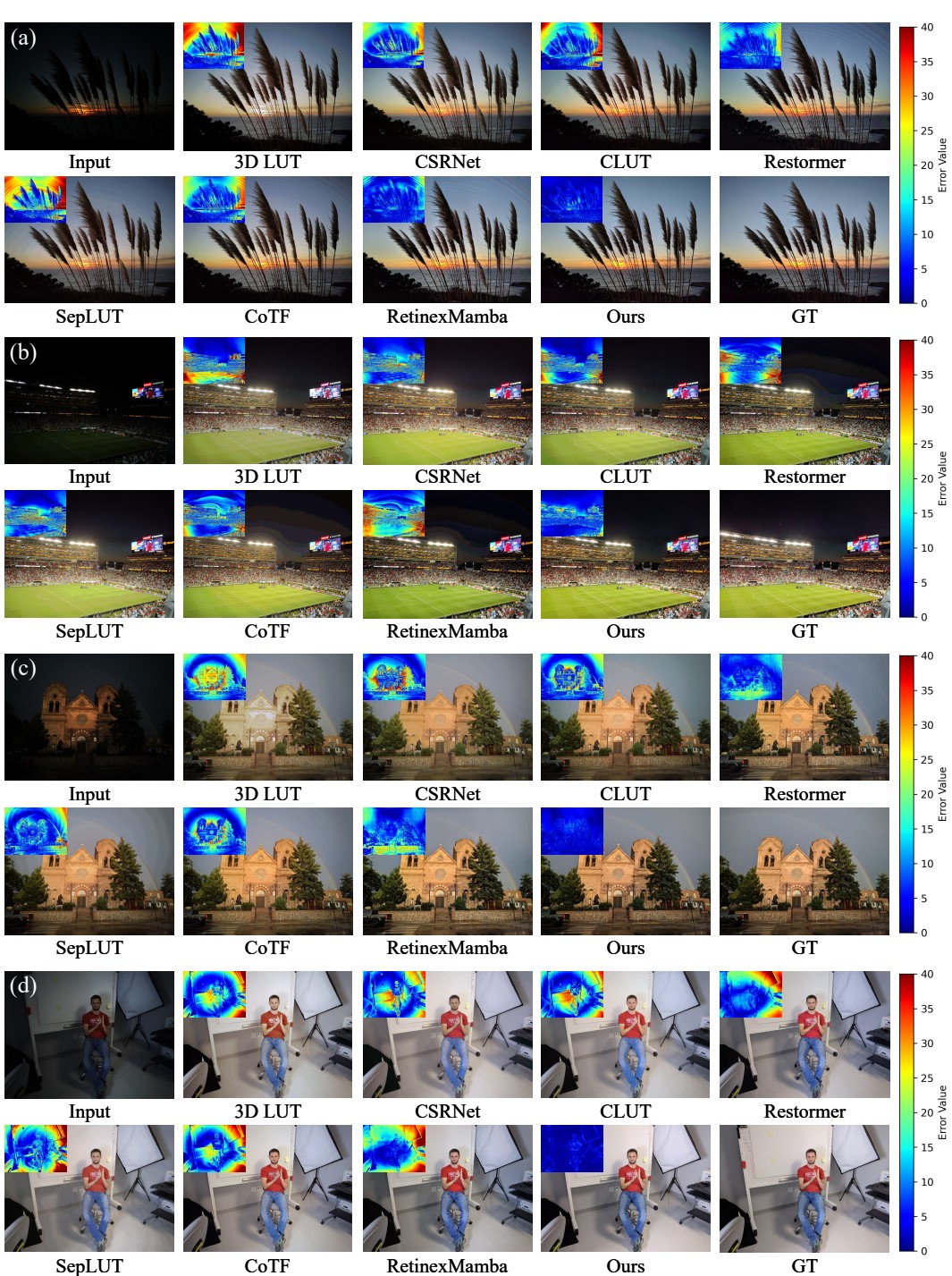

Figure 12: Visual comparisons between our LALet and the SOTA methods on the HDR+ dataset (Zoom-in for best view). The error maps in the upper left corner facilitate a more precise determination of performance differences.

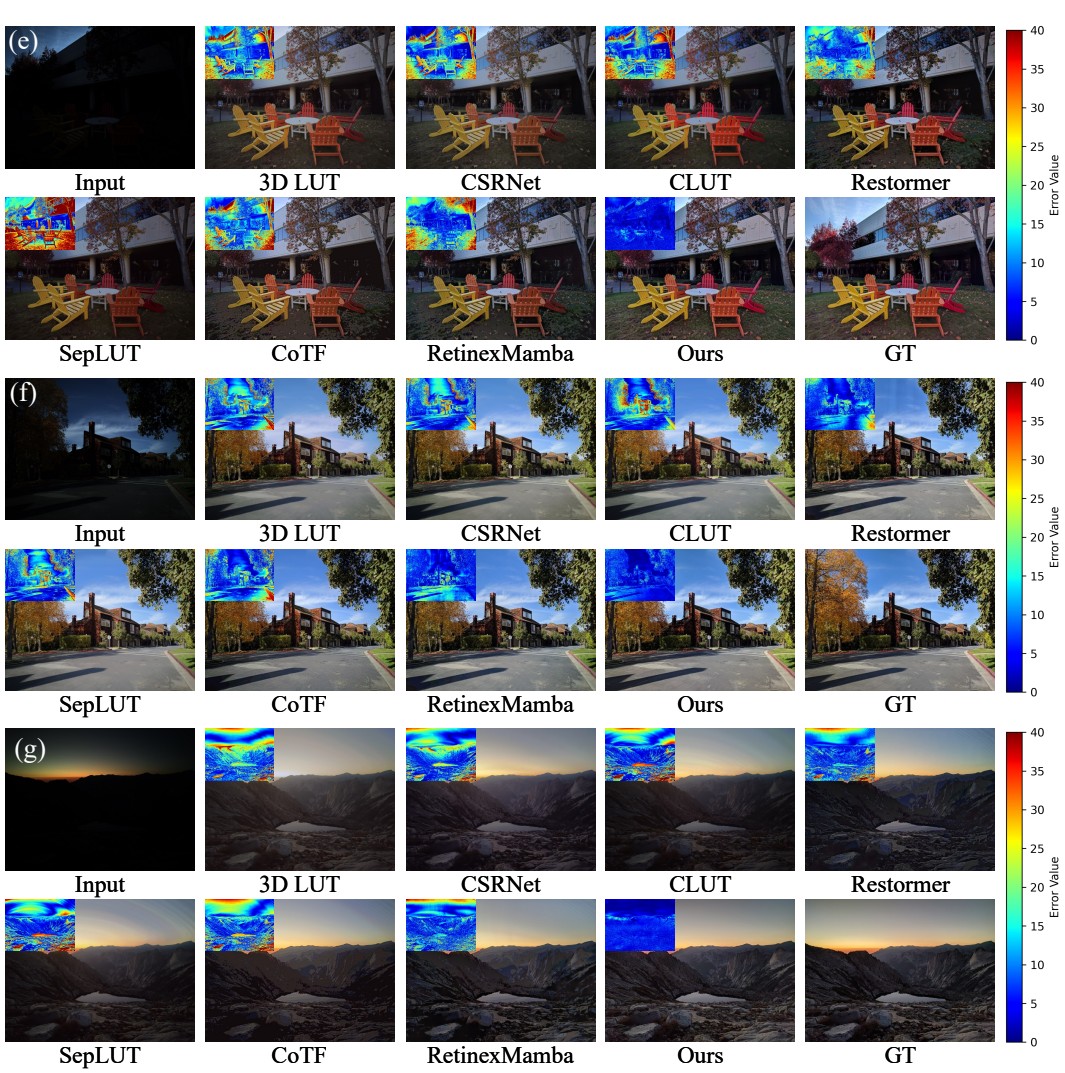

Figure 13: Visual comparisons between our LALet and the SOTA methods on the HDR+ dataset (Zoom-in for best view). The error maps in the upper left corner facilitate a more precise determination of performance differences.

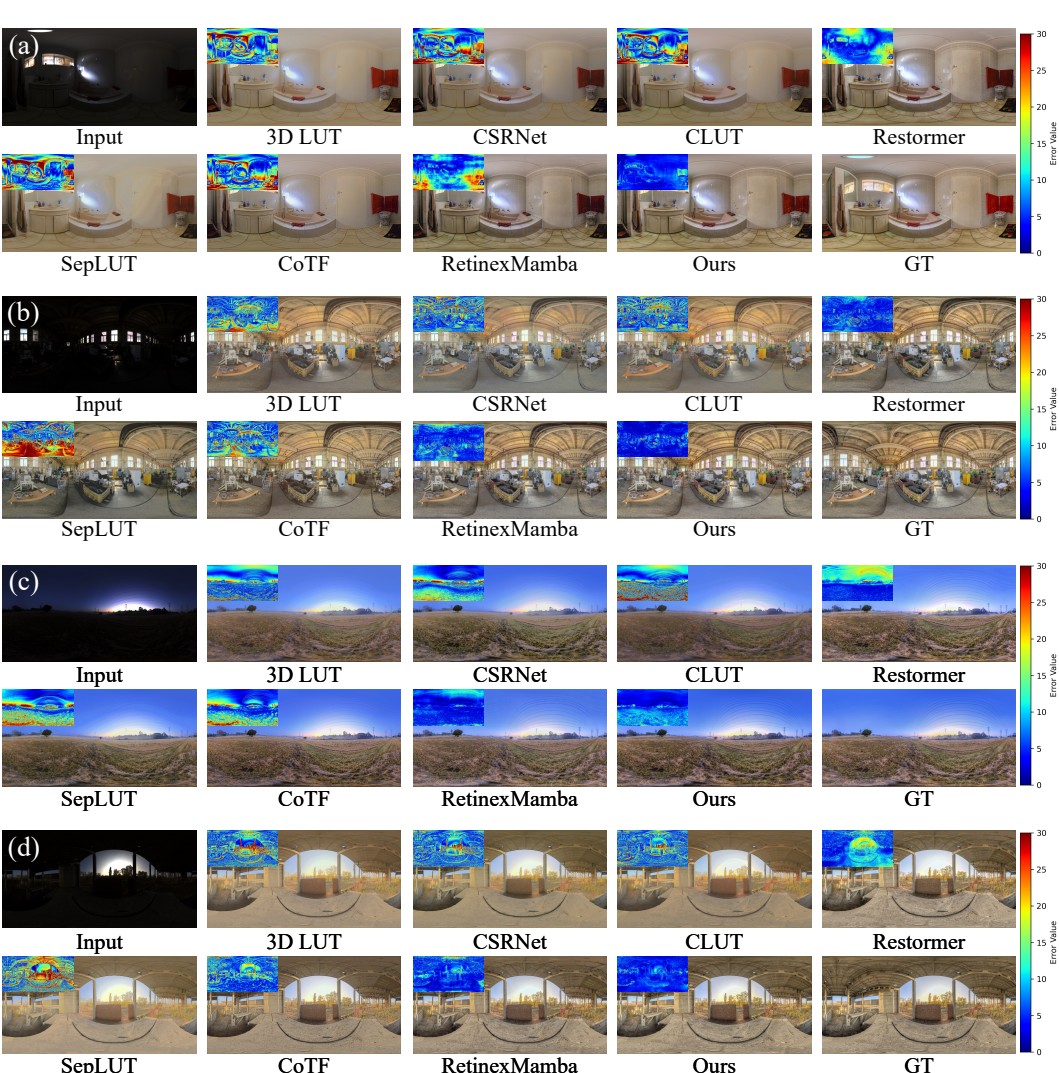

Figure 14: Visual comparisons between our LALNet and the SOTA methods on the HDRI Haven dataset (480p resolution).

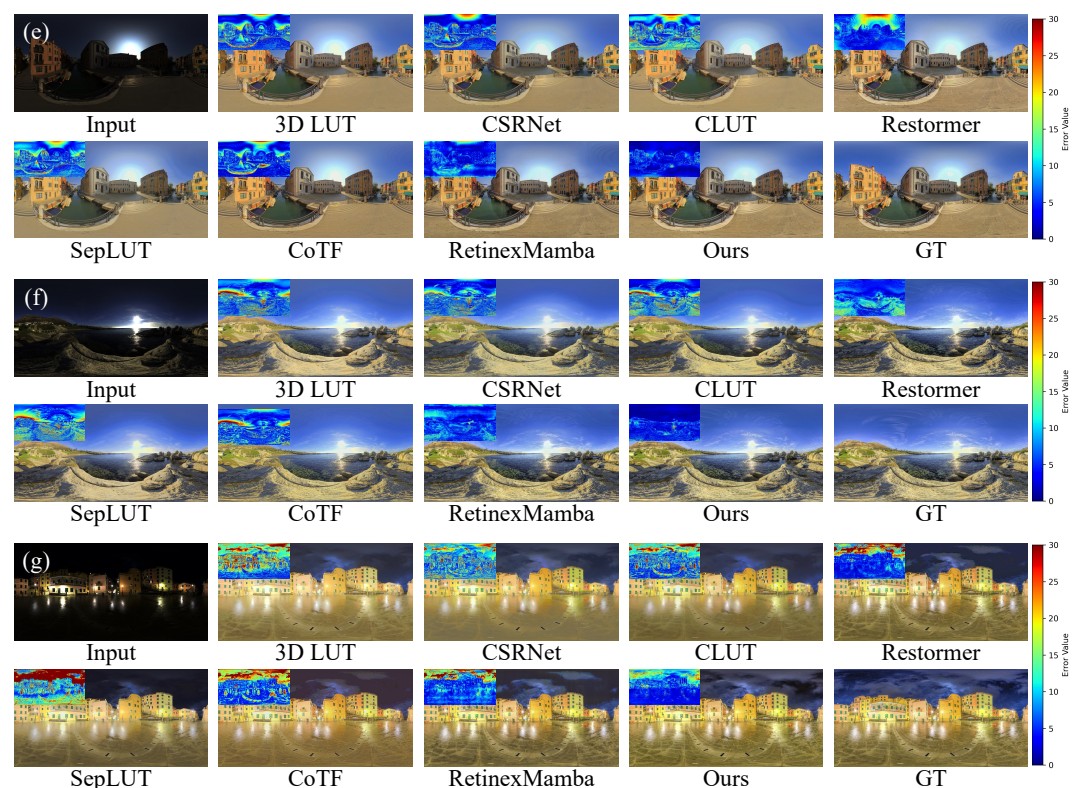

Figure 15: Visual comparisons between our LALNet and the SOTA methods on the HDRI Haven dataset (480p resolution).

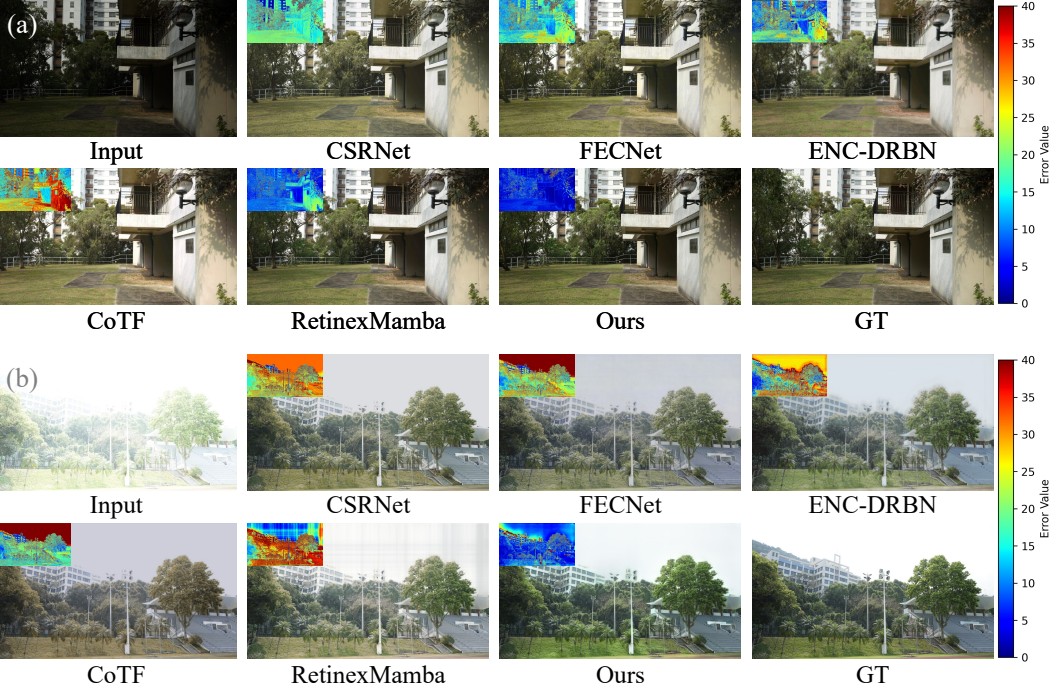

Figure 16: Visual comparisons between our LALNet and the SOTA methods on the SCIE dataset.

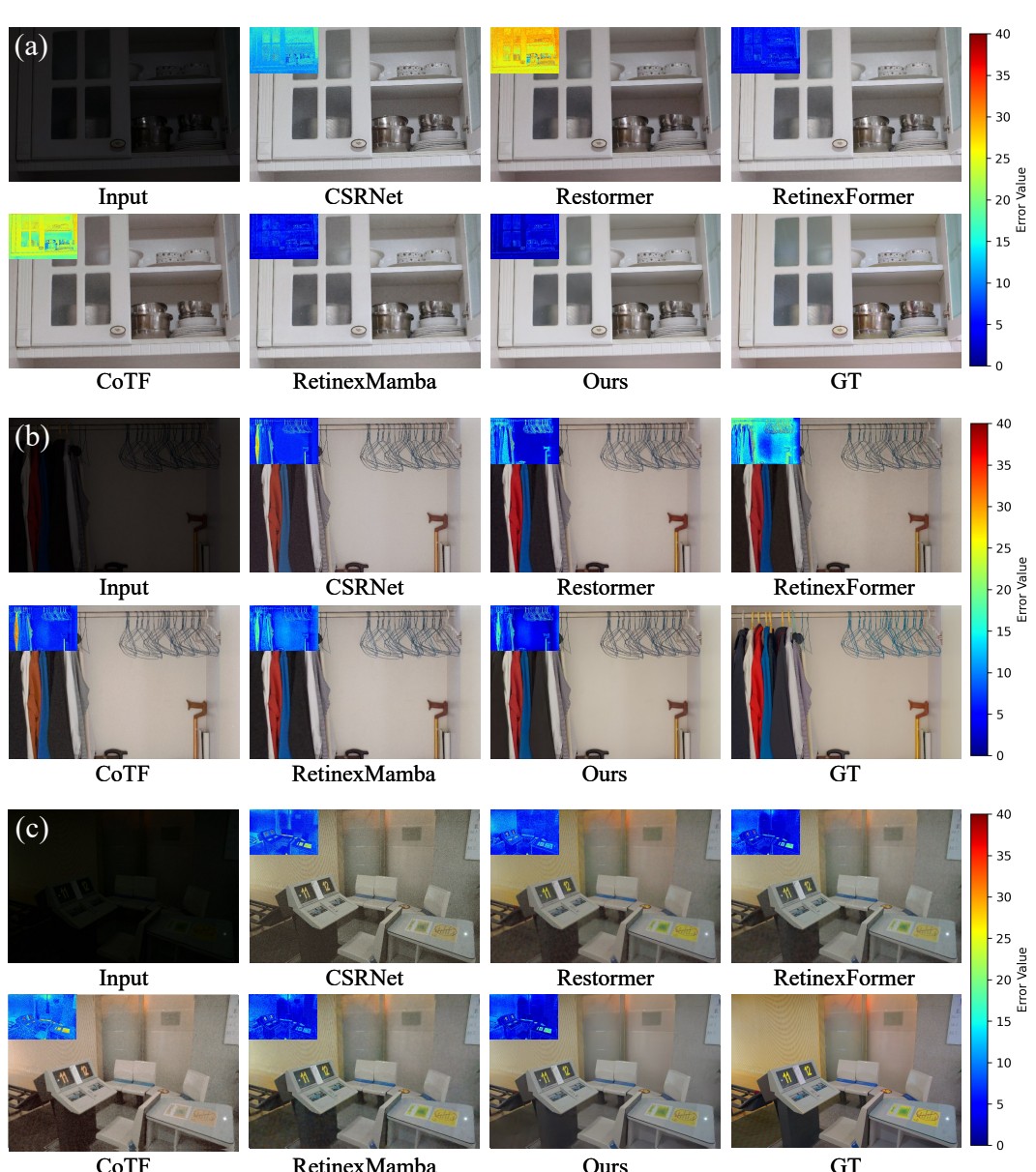

Figure 17: Visual comparisons between our LALNet and the SOTA methods on the LOLv1 dataset.

