# OpenReview forum: "Learning Adaptive Lighting via Channel-Aware Guidance"
_ICLR.cc/2025/Conference — ICLR 2025 Conference Withdrawn Submission_

### Official Review · Reviewer_pLv1 · 2024-10-22

**Soundness:** 3
**Presentation:** 3
**Contribution:** 3
**Rating:** 6
**Confidence:** 4

**Summary:**

The authors proposed a new method for lighting adaptation, which is inspired by the two key insights. Although such insights are not very new, the overall network structure is logically derived from them, making the network structure solid. A separate application of color-separated and color-mixed branches and their combination by LGA can be considered novel. Its effectiveness is justified by four different lighting-related tasks, which is also promising. The manuscript is well-written and well-organized.

**Strengths:**

The network structure consisting of the color-separated and color-mixed branches seems original and has enough technical contribution. Each branch is designed with persuasive supporting explanations. The proposed network shows SOTA performance on four different tasks, demonstrating significant margins from the second-best methods.

**Weaknesses:**

The authors' motivation is supported mainly by few examples in Figures 2 and 8. The authors are recommend to provide statistical analysis supporting the motivation rather than showing few chosen images.

I consider the experimental justification is somewhat biased. The authors used the results on the HDR+ dataset as a main performance evaluation, which is not a very widely used dataset. Almost all the baseline models in Table 1 are not developed for image retouching, making the comparison not fair enough. Consequently, as the ablation studies show, the model without DDCM and LGA, which does not contain technical contributions, already significantly outperforms the second-best method. I consider that the performance analysis and ablation studies should be conducted on the LLIE task using the LoL dataset.

**Questions:**

Almost all the baseline models in Table 1 are not developed for image retouching, making the comparison not fair enough. Consequently, as the ablation studies show, the model without DDCM and LGA, which does not contain technical contributions, already significantly outperforms the second-best method. I consider that the performance analysis and ablation studies should be conducted on the LLIE task using the LoL dataset.

---

> ### Author Response · Authors · 2024-11-21
> **Response to Reviewer pLv1**
>
> Thanks for your nice summary and positive comments. In the following, we give detailed responses to your comments.
>
> >Q1: The authors' motivation is supported mainly by few examples in Figures 2 and 8. The authors are recommend to provide statistical analysis supporting the motivation rather than showing few chosen images.
>
> A1: Thanks to the reviewer's suggestion, we have replaced Figure 2 and 8 in the updated manuscript to provide a statistical analysis of the energy distribution for each task. *Thanks again to the reviewers, these additions give broader empirical support to our motivation.*
>
> >Q2: I consider that the performance analysis and ablation studies should be conducted on the LLIE task using the LoL dataset.
>
> A2: Thank you for raising this concern. In the following, we give detailed responses to your comments.
>
> **Use of HDR+ Dataset:** We adopted the HDR+ dataset because it has been extensively used in recent works, including LLFLUT (NeurIPS 2023), which represents the current SOTA for this task. Moreover, established methods such as HDRNet (TOG 2017), CSRNet (ECCV 2020), LUT (TPAMI 2020), CLUT (MM 2022), and sLUT (ICCV 2021) have also utilized this dataset. This demonstrates that HDR+ is a well-accepted benchmark for evaluating performance in related tasks. While some baseline methods are not specifically designed for image retouching, they cover diverse approaches across related domains, showcasing the versatility of our framework in handling subtasks beyond its primary focus.
>
> **Additional Experiments on LoL Dataset:** To address the reviewer's concerns, we conducted additional experiments for the LLIE task using the LoL dataset. The results, presented in Table R6, further validate the effectiveness of our proposed modules (DDCM and LGA). These experiments reinforce our claims and demonstrate the robustness of our framework across different datasets and illumination-related tasks.
>
> Table R6: Ablation studies of key components on LOL dataset.
> | Variants | WFM | DDCM | LGA | PSNR↑ | SSIM↑ |
> |----------|-----|------|-----|-------|-------|
> | #1       | ✓   | ✗    | ✗    | 23.46 | 0.846 |
> | #2       | ✓   | ✓    | ✗    | 22.98 | 0.839 |
> | #3       | ✓   | ✗    | ✓    | 24.12 | 0.845 |
> | #4       | ✗   | ✓    | ✓    | 24.61 | 0.848 |
> | #5       | ✓   | ✓    | ✓    | 25.29 | 0.854 |
> |  |  |   |   |   |   |
>
> **Stability of training:**  We note that RetinexMamba and RetinexFormer report different results for the LoL dataset. We retrained both methods and observed that the training process is highly randomized and the results are prone to significant differences. Based on our retraining results, we chose RetinexMamba's reported results. However, this instability greatly affects the ablation experiments, therefore we chose the image retouching dataset (HDRPlus) for the main ablation experiments, as these tasks provide a more stable and reliable benchmark for evaluation.
>
> Once again, we sincerely thank the reviewers for their suggestions, which have greatly improved the completeness of our manuscript.

---

> > ### Comment · Reviewer_pLv1 · 2024-11-26
> >
> > Thanks for the revisions, which clarified most of my concerns. However, one of the critical concerns ("As the ablation studies show, the model without DDCM and LGA, which does not contain technical contributions, already significantly outperforms the second-best method.") is not addressed. I still consider that the performance comparison was not conducted with rigor.

---

> ### Author Response · Authors · 2024-11-28
> **Response to Reviewer’s Concerns on Ablation Study and Performance Comparison**
>
> We sincerely appreciate your continued feedback. We are grateful for your recognition of our revisions and would like to address the remaining concerns regarding the ablation study and the rigor of the performance comparison.
>
> **1. Performance Comparison Methods**
> The results reported in **Table 1** primarily come from **LLFLUT (NeurIPS 2023)**. We have not intentionally lowered the baseline performance. For other tasks, such as exposure correction (reported in **Table 2**), the results are derived from previously published papers **CoTF (CVPR 2024)**, ensuring a fair comparison with state-of-the-art methods. *Our codes will be released after the acceptance of this work.*
>
> **2. Additional Comparison Experiments**
> We acknowledge that methods based on Retinex theory, such as RetinexMamba and Retinexformer, perform suboptimally on tone mapping tasks due to inherent theoretical limitations. To address this, we have adopted **MambaIR (ECCV 2024)** as a new baseline. The results shown in **Table R7** demonstrate that even with a stronger baseline, our proposed model consistently outperforms existing methods.
>
> Table R7: Additional comparison experiments.
> | Variants |  #Params |  PSNR↑ | SSIM↑ |
> |----------|------------------|---------|-------|
> | MambaIR   | 4.31M   |  27.53 | 0.936 |
> | LALNet       | 2.87M   |  30.36 | 0.946 |
> |  |  |   |  |
>
> **3. Strength of the Baseline Model**
> The baseline model’s strong performance arises from several key components beyond DDCM and LGA, WFM, Learnable Differential Pyramid (LDP), Iterative Detail Enhancement (IDE) (detailed in Supplementary Material C), and the optimized backbone architecture. These components were specifically designed for light adaptation tasks and contribute significantly to the strength of the baseline. Although the efficacy of LDP and IDE was not emphasized in the main manuscript, they are new contributions introduced by our work. To verify their impact, we use LDP, IDE, and (CSRNet [ECCV2020]) to build a lightweight network, named LDPNet, which achieves comparable results to the state-of-the-art methods using only **89.52K** parameters (see **Table R8**).
>
> Table R8: Additional comparison experiments.
> | Variants |  #Params |  PSNR↑ | SSIM↑ |
> |----------|------------------|---------|-------|
> | Retinexformer   | 1.61M   |  26.20 | 0.910 |
> | RetinexMamba   | 4.59M   |  26.81 | 0.911 |
> | MambaIR   | 4.31M   |  27.53 | 0.936 |
> | LDPNet   | **0.089M**   |  26.52 | 0.914 |
> |  |  |   |  |
>
> Furthermore, the performance gains are further enhanced by the VSSM. As shown in Table R4, replacing VSSM with residual blocks or transformers results in a performance drop. These experiments demonstrate that the strength of our baseline is due to a combination of multiple factors.
>
> Table R4: Ablation study on the VSSM module.
> | Variants | Replaced Modules | #Params | FLOPs | PSNR↑ | SSIM↑ |
> |----------|------------------|---------|-------|-------|-------|
> | #1       | ResBlock         | 2.99M   | 7.13G | 29.77 | 0.9412 |
> | #2       | Self-Attention   | 2.25M   | 6.48G | 29.91 | 0.9380 |
> | #3       | Ours             | 2.87M   | 6.86G | 30.36 | 0.9458 |
> |  |  |   |  |  |  |
>
> **4. Core Technical Contributions**
> Although the baseline model is strong, it lacks the novel mechanisms central to our work—DDCM and LGA. These modules are essential for building a unified framework that can adapt to various lighting conditions. They effectively model the differing behaviors of color channels in both the spatial and frequency domains. Without these modules, the model would not achieve the same level of adaptive capability or generalization across different tasks.
>
> We hope this clarification, along with the additional analysis, addresses your concerns regarding the rigor of our performance comparison. Once again, we sincerely thank you for your thoughtful feedback, which has been invaluable in improving our manuscript.

---

> > ### Author Response · Authors · 2024-12-02
> > **Clarification on Baseline Performance and Technical Contributions**
> >
> > Dear Reviewer,
> >
> > Thank you for your thoughtful feedback and recognition of our revisions. We understand your concern regarding the strong baseline performance without DDCM and LGA, and we’d like to clarify further.
> >
> > **1. Baseline Strength:** The baseline’s performance arises from components like WFM, LDP, DDCM, and IDE, which are designed for light adaptation. These ensure a strong foundation but lack the novel adaptability provided by DDCM and LGA.
> >
> > **2. Rigor in Comparison:** We ensured fair comparisons by using published results from state-of-the-art methods (e.g., LLFLUT, CoTF) and included additional baselines (e.g., RetinexMamba, Retinexformer, MambaIR). Are there further experiments or clarifications you believe would enhance this rigor?
> >
> > **3. Core Contributions:** DDCM and LGA are essential to achieving adaptability across tasks. Without them, the model cannot generalize as effectively. We welcome any suggestions for additional analyses to highlight their impact.
> >
> > We appreciate your insights and are eager to refine our work further. Thank you for your time and consideration.
> >
> > Best regards,
> >
> > Authors of 1993

---

### Official Review · Reviewer_nP1E · 2024-11-01

**Soundness:** 3
**Presentation:** 3
**Contribution:** 3
**Rating:** 6
**Confidence:** 4

**Summary:**

This paper proposes a unified framework LALNet for light-related tasks, including HDR, Tone mapping, LLIE, Exposure correction. The argument is that different color channels demonstrate different lighting properties, and channel differences in time and frequency are also different. Based on that, a dual domain channel modulation is proposed with light guided attention for integration. Experiments are conducted respect to the four tasks, demonstrates some superiority according to results.

**Strengths:**

The proposed color channel properties regarding the lighting, is somewhat interesting, however, I still have some questions, see below

**Weaknesses:**

see the questions

**Questions:**

This paper is focused on lighting-related tasks, but I'm curious about what would happen if we switched to other color spaces, such as Lab or HSV, which separate color from luminance. Would it be easier to work with these color spaces compared to the RGB domain?

How about the retinex theory, e.g., intrinsic decomposition, that can seperate the illumination from textures. Would it be easilier than working in the rgb domain where lighting mixed with colors, as discussed in the paper, rgb space demonstrates different channel porperties in color and frequency

For the evaluation, authors are suggested to compare with the SOTA methods respect to each of the task. Currently, some of the SOTA methods are omitted.

For some visual comparisons, it is hard to notice the differences when looking at the color images.

---

> ### Author Response · Authors · 2024-11-21
> **Response to Reviewer nP1E**
>
> Thanks for your nice summary and positive comments. In the following, we give detailed responses to your comments.
> >Q1: This paper is focused on lighting-related tasks, but I'm curious about what would happen if we switched to other color spaces, such as Lab or HSV, which separate color from luminance. Would it be easier to work with these color spaces compared to the RGB domain?
>
> A1: Thank you for the question.  In the following, we give detailed responses to your comments.
>
> **Task Adaptability:** There have been works based on the HSV space, which keep the H (hue) channel fixed and manipulate the tone mapping in the S (saturation) and V (luminance) channels.
> While the HSV and Lab spaces are suitable for tasks that require independent manipulation of color properties, such as color tuning. However, for tasks such as exposure correction and low-light enhancement that require luminance and high-frequency information, the luminance (e.g., L or V channels) is obtained through nonlinear transformations, which may result in loss of or distortion of luminance details. Therefore, RGB space is more widely applicable.
>
> **Lighting Properties:** Common image sensors are usually based on the Bayer pattern (RGGB), so the RGB domain more faithfully reflects the lighting and color information of the raw image. Essentially, the unique light-related properties we observe are due to the different mask colors of the image sensor. Therefore, for lighting related tasks, the RGB color space is more advantageous.
>
> **Future Directions:** Nevertheless, we agree that exploring other color spaces may be an interesting direction for future work. Therefore, we will add this section to the related work and discussions to promote the color tone community.
>
> >Q2: How about the retinex theory?
>
> A2: Thank you for your insightful comments and questions. We give detailed responses to your comments.
>
> **Limitations:** Retinex-based methods are based on the theory of separated illumination and reflection, but they usually assume smooth and uniform lighting conditions, which may not hold in realistic scenes involving complex lighting variations. In addition, these methods typically work in luminance or reflection space, where high-frequency details may be distorted during decomposition. While intrinsic decomposition can simplify some aspects of lighting modeling, RGB-based methods offer greater flexibility in capturing real-world complexity. In fact, we have discussed the Retinex method in the introduction as well as in related work.
>
> **Increased discussion:** We will include a discussion of retinex theory in the related work section.
>
> >Q3: For the evaluation, authors are suggested to compare with the SOTA methods respect to each of the task. Currently, some of the SOTA methods are omitted.
>
> A3: Yes, we have ignored some diffusion-based low light enhancement methods. Also, some new SOTA methods appeared at the submission stage. We will add this in the new version of the manuscript and discuss the computational cost advantages of our method.
>
> >Q4: For some visual comparisons, it is hard to notice the differences when looking at the color images.
>
> A4: Thank you for your feedback. We understand that subtle differences in visual comparisons may not always be obvious, especially when evaluating lighting and tonal tuning.
>
> To address this issue, we will be adding more visualizations and adding slidable the [Web page comparisons](https://xxxxxx2025.github.io/LALNet/). With the webpage slider comparison, you can observe significant visual differences.
>
> We sincerely thank the reviewers for their comments, which greatly improved the integrity of our manuscript.

---

### Official Review · Reviewer_gdPq · 2024-11-02

**Soundness:** 2
**Presentation:** 3
**Contribution:** 3
**Rating:** 5
**Confidence:** 4

**Summary:**

The paper presents Learning Adaptive Lighting Network, a unified framework for multiple light-related image enhancement tasks, including image retouching, tone mapping, low-light enhancement, and exposure correction. The authors propose channel-aware guidance to handle the unique properties of each color channel, integrating color-separated and color-mixed features through a Dual Domain Channel Modulation and Light Guided Attention. Experimental results show that LALNet achieves state-of-the-art performance on benchmark datasets, outperforming existing methods both in accuracy and computational efficiency.

**Strengths:**

- The paper presents a novel approach by leveraging color-separated and color-mixed features to address multiple light-related tasks within a single model, which reflects a strong methodological innovation.
- The results across four different tasks and datasets, with quantitative comparisons, convincingly support the effectiveness of LALNet, showing substantial improvements over state-of-the-art methods.
- The distinction between time and frequency domain representations of different color channels is well-grounded.

**Weaknesses:**

- The paper does not clarify whether LALNet shares parameters between models evaluated across different tasks, leaving it unclear if LALNet operates as a truly unified model. This aspect is crucial to evaluate the generalizability of LALNet as a single framework capable of adapting to diverse lighting tasks.
- The main contributions rely heavily on combining established well-known methods (e.g., wavelet-based modulation, vision state-space models, and channel-wise attention) with minimal modifications. Given recent work in similar domains, the marginal improvements over these methods do not provide a substantial contribution to the field.
- The paper could benefit from a clearer architectural diagram to visually depict the relationships between DDCM, LGA, and wavelet modulation.
- Terminology like “adaptive lighting” could be further clarified to distinguish it from traditional exposure correction or low-light enhancement.
- Formatting inconsistencies in equations (e.g., missing commas or misaligned subscripts) detract slightly from readability.

**Questions:**

- Does LALNet share parameters across tasks, or is a separate model trained and optimized for each task? How do you address cross-task generalization if parameters are not shared?
- How does LALNet perform on new or more diverse lighting conditions, such as HDR imaging in natural outdoor scenes?
- Could you clarify the contribution of VSSM versus traditional convolution-based approaches? Specifically, how much does VSSM contribute to cross-channel consistency?

---

> ### Author Response · Authors · 2024-11-21
> **Response to Reviewer gdPq**
>
> Thanks for your nice summary and positive comments. In the following, we give detailed responses to your comments.
>
> >Q1: Does LALNet share parameters across tasks, or is a separate model trained and optimized for each task? How do you address cross-task generalization if parameters are not shared?
>
> A1: LALNet is a unified framework where models for each task are individually training and optimized without sharing parameters.
> We have performed generalizability experiments on the same task, where we test additional HDR images and video data using a model trained on HDRI Haven data. As shown in Table R2, the results show that our model achieves the best generalization.
>
> Table R2: Validating generalization on third-party datasets include HDR Survey and UVTM video datasets.
> | Datasets | Metrics | HDRNet | CSRNet | 3D LUT | CLUT | SepLUT | IVTMNet | CoTF | Ours |
> |----------|---------|--------------------------------|-----------------------------|---------------------------|-----------------------|------------------------------|----------------------------------|----------------------|-----|
> | HDR Survey | TMQI | 0.8641 | 0.8439 | 0.8165 | 0.8140 | 0.8085 | 0.9160 | 0.8612 | **0.9292** |
> | UVTM | TMQI | 0.8281 | 0.8973 | 0.8787 | 0.8799 | 0.8629 | 0.8991 | 0.9006 | **0.9576** |
> |   |    |  |   |    |   |   |   |    |   |
>
> >Q2: The main contributions rely heavily on combining established well-known methods.
>
> A2: We thank the reviewers for their observations and wish to clarify our contribution. It is important to clarify that established well-known methods are not the primary contributions of our work, and the performance gains are not derived from it. As shown in Table R3, DDCM and LGA are the main contributors to the performance gain. Moreover, we perform ablation study on the VSSM module, and the results demonstrate that VSSM is not a main performance gain compared to DCCM and LGA, see in Table R4.
>
> Table R3: Ablation studies of key components.
> | Variants | WFM | DDCM | LGA | PSNR↑ | SSIM↑ |
> |----------|-----|------|-----|-------|-------|
> | #1       | ✓   | ✗    | ✗    | 29.11 | 0.933 |
> | #2       | ✓   | ✓    | ✗    | 29.58 | 0.935 |
> | #3       | ✓   | ✗    | ✓    | 30.01 | 0.942 |
> | #4       | ✗   | ✓    | ✓    | 30.05 | 0.942 |
> | #5       | ✓   | ✓    | ✓    | 30.36 | 0.946 |
> |  |  |   |  |  |  |
>
> Table R4: Ablation study on the VSSM module.
> | Variants | Replaced Modules | #Params | FLOPs | PSNR↑ | SSIM↑ | TMQI↑ | LPIPS↓ | ΔE↓ |
> |----------|------------------|---------|-------|-------|-------|-------|--------|------|
> | #1       | ResBlock         | 2.99M   | 7.13G | 29.77 | 0.9412 | 0.8781 | 0.0291 | 4.760 |
> | #2       | Self-Attention   | 2.25M   | 6.48G | 29.91 | 0.9380 | 0.8801 | 0.0297 | 4.872 |
> | #3       | Ours             | 2.87M   | 6.86G | **30.36** | **0.9458** | **0.8883** | **0.0261** | **4.483** |
> |  |  |   |  |  |  | |  |  |
>
> Specifically, our contribution is driven by the unique insight that different color channels exhibit different light properties that vary in the time and frequency domains. We incorporated this insight into a unified framework, designing **Dual-Domain Channel Modulation (DDCM)** and **Light Guided Attention (LGA)** based on these insights.
> Please allow us to once again highlight my main contributions:
> - In this paper, we propose a unified light adaptation framework inspired by the common light property, namely the Learning Adaptive Lighting Network (LALNet).
> - We introduce the Dual Domain Channel Modulation to capture the light differences of different color channels and combine them with the traditional color-mixed features by Light Guided Attention.
>
> >Q3: The paper could benefit from a clearer architectural diagram to visually depict the relationships between DDCM, LGA, and wavelet modulation.
>
> A3: In order to better describe the relationship between DDCM, LGA and wavelet modulation, we have updated the architecture diagram and manuscript. We add math notation and update this part of the description. Specifically, for first LAG module, we input the $ X^3_{LF} $ to the DDCM and WFM, respectively, to obtain $ F_{cm} $ and $ F_{cs} $, described as:
> $$
> F_{cm} = WFM(X^3_{LF}),        F_{cs} = DDCM(X^3_{LF}).
> $$
> Then, $ F_{cm} $ and $ F_{cs} $ are input to the group convolution and VSSM modules, respectively, to obtain $ F^1_{cm} $ and $ F^1_{cs} $, which are described as follows:
> $$
> F^1_{cm} = GConv(F_{cm}),     F^1_{cs} = VSSM(F_{cs}).
> $$
> Finally, $ F^1_{cm} $ and $ F^1_{cs} $ are inputted into LGA to obtain the light adaptive feature $ F^1_{la} $, which is described as:
> $$
> F^1_{la} =LGA(F^1_{cm}, F^1_{cs}).
> $$
> We hope that the updated diagram will ensure that the method is easier for reviewers to understand. Thank you for pointing out this opportunity for improvement.

---

> ### Author Response · Authors · 2024-11-21
> **Response to Reviewer gdPq**
>
> >Q4: Terminology like “adaptive lighting” could be further clarified to distinguish it from traditional exposure correction or low-light enhancement.
>
> A4: The performance of the proposed model is evaluated on four light-related image-enhancement tasks, namely, (1) image retouching (IR), (2) HDR tone mapping (TM), (3) exposure correction (EC), and (4) low-light image enhancement (LLIE). The main characteristics of each task are summarized in Table R5, which shows that the common operator of all four tasks is light adaptation. Therefore, we refer to them collectively as light adaptation.
>
> Table R5: Quantitative results of image retouching methods.
> | Task | Light  | Objectives |
> |--------|--------------|---------------|
> | IR | over/under exposure, poor contrast |  Contrast enhancement and lighting inharmonious |
> | TM | High dynamic range | Dynamic range compression and detail preservation  |
> | EC | Both over- and underexposure  |  Light correction and detail enhancement |
> | LLIE | Darkness |  Lighting and denoising |
> |   |    |  |
>
> >Q5: Formatting inconsistencies in equations.
>
> A5: I thank the reviewers for their careful review. We will address these typos in the revised version and improve our manuscript.
>
> >Q6: How does LALNet perform on new or more diverse lighting conditions, such as HDR imaging in natural outdoor scenes?
>
> A6: We validate our model on non-homologous third-party image and video HDR datasets that contain a wide range of lighting conditions. As shown in Table 2, our model is far superior to existing methods. We show two challenging nighttime lighting scenarios, and the visualization results shown in Figure 11 (in Appendix) further confirm our superiority.
>
> >Q7: Could you clarify the contribution of VSSM versus traditional convolution-based approaches?
>
> A7: In fact, we have conducted this experiment, see Table R4 and Appendix (Table 7). We replace VSSM with self-attention and residual block, and the results show that VSSM is more effective for global light feature extraction than Self-Attention and residual block.
>
> Essentially, based on the effectiveness of wavelets and Mamba in capturing global features, we utilize them to extract global channel mixed features to ensure the basic performance of the overall framework.
>
> Thanks again for your valuable comments.

---

### Official Review · Reviewer_vkeS · 2024-11-03

**Soundness:** 2
**Presentation:** 2
**Contribution:** 1
**Rating:** 3
**Confidence:** 5

**Summary:**

This paper proposes a framework for learning adaptive lighting with light property guidance. First, the authors propose DDCM for extracting color-separated features and capturing the light difference across channels. Then, the LGA utilizes color-separated features to guide color-mixed features for adaptive lighting, achieving color consistency and color balance. Extensive experiments demonstrate that the proposed method outperforms state-of-the-art methods on various light-related benchmarks.

**Strengths:**

1, The proposed method achieves state-of-the-art performance.
2, The method is easy to follow.

**Weaknesses:**

1, The novelty is limited. The model mainly combines existing modules to form the proposed method, e.g., wavelet, mamba, etc.

2, The authors need to improve the writing. The current version cannot satisfy the standard of ICLR.

3, The authors should provide the comparisons on real-world data as well as the non-reference metrics such as NIQE, PI, PIQE, etc.

4, The motivation of the paper is not insightful enough.

**Questions:**

See Weaknesses.

---

> ### Author Response · Authors · 2024-11-21
> **Response to Reviewer vkeS**
>
> Thank you for your thorough review and for taking the time to consider our response. We would like to further clarify the technical novelty and contributions of our work.
>
> >Q1: Response to Limited Novelty Concerns.
>
> A1: We appreciate your concern regarding the novelty of our work. We would like to clarify that while wavelets and Mamba are incorporated into our framework, they are not the core contributions of this framework. Rather, they serve as components within a unified framework, leveraging their effectiveness in capturing global features to help extract global channel-mixed information. To ensure clarity, we have conducted ablation studies on these modules (wavelet and Mamba), which are detailed in the Ablation Study and Appendix sections.
>
> Our primary contribution lies in the novel insight that different color channels exhibit distinct lighting properties, and these differences manifest differently in both the time and frequency domains. Building on this observation, we introduced a Dual Domain Channel Modulation (DDCM) module, which extracts these channel-specific differences from both domains. Additionally, we developed a Light Guided Attention (LGA) mechanism to integrate these extracted differences into the adaptive lighting process. This DDCM and LGA enhance the model’s ability to perceive and adapt to channel luminance variations, a key step for achieving unified light adaptation.
>
> Please allow us to once again highlight my main contributions:
> - In this paper, we propose a unified light adaptation framework inspired by the common light property, namely the Learning Adaptive Lighting Network (LALNet).
> - We introduce the Dual Domain Channel Modulation to capture the light differences of different color channels and combine them with the traditional color-mixed features by Light Guided Attention.
> - Extensive experiments on four representative light-related tasks show that LALNet significantly outperforms state-of-the-art methods in benchmarking and that our method requires fewer computational resources.
>
> >Q2: The authors need to improve the writing.
>
> A2: We appreciate the reviewer's comment highlighting the need for improvements in writing.
> - Revision and Refinement: We have thoroughly revised the manuscript to improve clarity, conciseness, and overall readability.
> - Consistency and Formatting: We have carefully reviewed the paper to ensure consistent terminology, proper citation formatting, and adherence to the ICLR style guidelines.
> - Collaborative Proofreading: The updated manuscript has been proofread by multiple collaborators to ensure grammatical accuracy and alignment with the high standards of ICLR.

---

> ### Author Response · Authors · 2024-11-21
> **Response to Reviewer vkeS**
>
> >Q3: Comparison of real-world data and non-referenced metrics
>
> A3: Thanks for the suggestion . We add more non-reference metrics (TMQI, NIQE, MUSIQ). Meanwhile, we test our model on more real-world datasets (HDR Survey and video dataset UVTM), which is trained on the HDRI Haven dataset.
> The experimental results are shown in Table R1 and Table R2, which show that our method is far superior to existing methods and also shows that our framework has good generalization performance.
>
> Table R1: Quantitative results of image retouching methods.
> | Method | #Params  | PSNR↑ | SSIM↑ | TMQI↑ | LPIPS↓ | ΔE↓ | NIQE↓ | MUSIQ↑ |
> |--------|---------|-------|-------|-------|--------|------|-------|--------|
> | UPE | 999K |  23.33 | 0.852 | 0.856 | 0.150 | 7.68 | 12.75 | 66.98 |
> | HDRNet | 482K |  24.15 | 0.845 | 0.877 | 0.110 | 7.15 | 10.47 | 68.73 |
> | CSRNet | 37K |  23.72 | 0.864 | 0.884 | 0.104 | 6.67 | 10.99 | 67.82 |
> | DeepLPF | 1.72M |  25.73 | 0.902 | 0.877 | 0.073 | 6.05 | 10.35 | 70.02 |
> | LUT | 592K |  23.29 | 0.855 | 0.882 | 0.117 | 7.16 | 11.36 | 67.67 |
> | CLUT | 952K |  26.05 | 0.892 | 0.886 | 0.088 | 5.57 | 11.19 | 67.39 |
> | LPTN | 616K |  24.80 | 0.884 | 0.885 | 0.087 | 8.38 | 12.44 | 67.99 |
> | sLUT | 4.52M |  26.13 | 0.901 | / | 0.069 | 5.34 | / | / |
> | SepLUT | 120K |  22.71 | 0.833 | 0.879 | 0.093 | 8.62 | 12.26 | 67.89 |
> | Restormer | 26.1M |  25.93 | 0.900 | 0.883 | 0.050 | 6.59 | 10.49 | 68.92 |
> | LLFLUT | 731K |  26.62 | 0.907 | / | 0.063 | 5.31 | / | / |
> | CoTF | 310K |  23.78 | 0.882 | 0.876 | 0.072 | 7.76 | 11.54 | 68.07 |
> | Retinexformer | 1.61M |  26.20 | 0.910 | 0.879 | 0.046 | 6.14 | 10.75 | 68.93 |
> | RetinexMamba | 4.59M |  26.81 | 0.911 | 0.880 | 0.047 | 5.89 | 10.52 | 69.02 |
> | **LALNet** | **2.87M** |  **30.36** | **0.946** | **0.888** | **0.026** | **4.48** | **9.87** | **70.29** |
> |   |    |  |   |    |   |
>
> Table R2: Validating generalization on third-party datasets include HDR Survey and UVTM video datasets.
> | Datasets | Metrics | HDRNet | CSRNet | 3D LUT | CLUT | SepLUT | IVTMNet | CoTF | Ours |
> |----------|---------|--------------------------------|-----------------------------|---------------------------|-----------------------|------------------------------|----------------------------------|----------------------|-----|
> | HDR Survey | TMQI | 0.8641 | 0.8439 | 0.8165 | 0.8140 | 0.8085 | 0.9160 | 0.8612 | **0.9292** |
> | UVTM | TMQI | 0.8281 | 0.8973 | 0.8787 | 0.8799 | 0.8629 | 0.8991 | 0.9006 | **0.9576** |
> |   |    |  |   |    |   |   |   |    |   |
>
> >Q4: The motivation of the paper is not insightful enough.
>
> A4: We sincerely thank the reviewer for their feedback. We appreciate the opportunity to further clarify the motivation behind our work:
>
> **Lack of Task Generalization in Existing Methods:** Existing methods for light-related tasks such as tone mapping and low-light enhancement are often tailored to individual tasks, resulting in suboptimal performance across multiple scenarios. These methods typically fail to account for the common properties shared across different lighting-related tasks, which limits their generalizability. As a result, many frameworks are either overly specialized or inefficient when faced with multiple tasks. This leads to performance inconsistencies, especially when methods designed for specific tasks are applied to others.
>
> **Unified Framework Inspired by Common Light Properties:** Our motivation is rooted in the observation that despite the diverse nature of light-related tasks, there are key shared properties. In particular, we found that:
>
> - Different color channels exhibit distinct light properties, which affect how lighting and textures are perceived.
> - These channel differences manifest differently in both time and frequency domains, which further complicates the task of adaptive lighting adjustment.
>
> By analyzing these shared light properties across multiple tasks, we design a unified framework that adapts to different lighting conditions more effectively than previous methods that focus on individual tasks.
>
> **Innovative Design of DDCM and LGA:** Based on these insights, we propose two novel modules: Dual Domain Channel Modulation (DDCM) and Light Guided Attention (LGA). These modules specifically address the observed channel differences, extracting and integrating them from both time and frequency domains. The result is a unified lighting adaptation framework that not only captures subtle differences between color channels but also ensures adaptive adjustments across various light-related tasks. This leads to enhanced visual consistency and color balance across different lighting conditions.
>
> We hope this clarification offers a clearer perspective on the motivation behind our work.
>
> We would like to thank you once again for your thoughtful review and hope that this response addresses your concerns about the motivation behind our research.

---

### Author Response · Authors · 2024-11-25
**General Response: Contributions and New Experiments**

We sincerely thank all the reviewers for their insightful and valuable comments! Overall, we are encouraged that they find that:

1. The proposed model is a novel image enhancement framework with production-ready quality. (Reviewers gdPq, nP1E, pLv1)

2. Each branch of the proposed design is well-supported with convincing explanations and comprehensive experiments. (Reviewers pLv1)

3. The consideration of color channel properties for illumination is a novel approach. (Reviewer nP1E)

And we also want to emphasize our contributions: Through in-depth exploration of light attributes, we have designed light attribute-based feature extraction modules to enhance light adaptation. LALNet unifies light-related tasks within a single framework for the first time. This work establishes a foundational framework for the advancement of multiple communities and provides a prerequisite for future All-in-One models.

In response to the reviewers’ feedback, we have made the following revisions to improve the manuscript:

**Abstract:** We provided an anonymous demo link to clearly demonstrate the performance differences between the models.

**Introduction:** We included statistical analyses to support the motivation, updating Figures 2 and 7.

**Methods:** We explicitly presented the motivation behind our approach, updating Section 2.1 Motivation.

**Architectural Diagram:** We added mathematical representations to the diagram and supplemented it with detailed mathematical descriptions, updating Figure 3 and the description.

**Experiments:** Additional no-reference metrics are added to enhance experimental validation, NIQE and MUSIQ metrics are added.

**Appendix A:** We added a discussion on HSV and Retinex to address reviewer concerns.

**Appendix D:** We tested additional real-world HDR image and video datasets and provide a comprehensive visual comparison, see Table 6 and Figure 11.

We carefully addressed all issues raised by reviewers and highlighted these changes in yellow for easy reference.

Next, we address each reviewer's detailed concerns point by point. We sincerely thank all reviewers for their recognition of our work and the valuable suggestions provided. Discussions are always welcome. Thank you!

---

### Note · Authors · 2025-01-23

I have read and agree with the venue's withdrawal policy on behalf of myself and my co-authors.